# Selective prebiotic conversion of pyrimidine and purine anhydronucleosides into Watson-Crick base-pairing *arabino*-furanosyl nucleosides in water

Samuel J. Roberts[1], Rafał Szabla[2,3], Zoe R. Todd[4], Shaun Stairs[1], Dejan-Krešimir Bučar[1], Jiří Šponer[3], Dimitar D. Sasselov[4] & Matthew W. Powner [1]

Prebiotic nucleotide synthesis is crucial to understanding the origins of life on Earth. There are numerous candidates for life's first nucleic acid, however, currently no prebiotic method to selectively and concurrently synthesise the canonical Watson–Crick base-pairing pyrimidine (C, U) and purine (A, G) nucleosides exists for any genetic polymer. Here, we demonstrate the divergent prebiotic synthesis of arabinonucleic acid (ANA) nucleosides. The complete set of canonical nucleosides is delivered from one reaction sequence, with regiospecific glycosidation and complete furanosyl selectivity. We observe photochemical 8-mercaptopurine reduction is efficient for the canonical purines (A, G), but not the non-canonical purine inosine (I). Our results demonstrate that synthesis of ANA may have been facile under conditions that comply with plausible geochemical environments on early Earth and, given that ANA is capable of encoding RNA/DNA compatible information and evolving to yield catalytic ANA-zymes, ANA may have played a critical role during the origins of life.

[1] Department of Chemistry, University College London, 20 Gordon Street, London WC1H 0AJ, UK. [2] Institute of Physics, Polish Academy of Sciences, Al. Lotników 32/46, PL-02668 Warsaw, Poland. [3] Institute of Biophysics of the Czech Academy of Sciences, Královopolská 135, 61265 Brno, Czech Republic. [4] Harvard-Smithsonian Center for Astrophysics, Department of Astronomy, Harvard University, 60 Garden Street, Cambridge, MA 02138, USA. Correspondence and requests for materials should be addressed to M.W.P. (email: matthew.powner@ucl.ac.uk)

The synthesis of the complete set of canonical Watson–Crick base-pairing nucleosides [adenosine (A), cytidine (C), guanosine (G) and uridine (U)] under conditions that do not violate the accepted plausible geochemical environments on early Earth is an essential step towards elucidating the origins of life on Earth[1–3]. However, while many 'plausible' nucleoside candidates have been suggested to have played a role at the origins of life[4–6], the concurrent prebiotic synthesis of a complete set of nucleoside monomers remains an unresolved challenge for any of the proposed genetic polymers [e.g., ribonucleic acid (RNA), arabinonucleic acid (ANA), threonucleic acid (TNA) and pyranosyl-ribonucleic acid (pRNA)][3,5,7–24]. Accordingly, we set out to elucidate chemical reactions that could address this problem. We have previously reported a prebiotic synthesis of pyrimidine ribonucleotides 1C and 1U[16]. More recently, we reported the divergent synthesis of 1C, 1U and 8-oxo-purine ribonucleotides 2A and 2I (Fig. 1, red arrows)[22]. However, no divergent prebiotic synthesis of pyrimidine and purine nucleoside monomers bearing the canonical Watson–Crick base-pairing nucleobases has yet been elucidated[7,21–23].

ANA displays many properties that make it an attractive candidate for the first genetic polymer of life. ANAs can equilibrate between helix and stem-loop structures, which mimic DNA and RNA, respectively[25]. ANA can form a complementary Watson–Crick base-paired duplex with RNA[26,27], and can be readily transcribed (from DNA) and reverse transcribed (to DNA)[28]. Additionally, Holliger and co-workers recently evolved catalytic ANA-zymes that can achieve RNA phosphodiester cleavage[6]. Notably, the ANA phosphodiester backbone is also far more resistant to hydrolysis than its RNA analogue[26].

Anhydrocytidine (3C), a key intermediate in our previously reported prebiotic pyrimidine synthesis[16], undergoes hydrolysis at near neutral pH ($\geq 6.5$)[22] to quantitatively afford arabino-cytidine (ara-4C; Fig. 1, blue arrow). This facile hydrolysis suggests that a simple prebiotic synthesis of arabino-nucleotides may be achievable. Importantly, the synthesis of a complete set of arabinosides requires differential reactivity between the purine and pyrimidine precursors 3A/3G and 3C (Fig. 1),

respectively[16,22]. Pyrimidine ara-4C can be accessed by direct hydrolysis of 3C, whereas hydrolysis of 3A and 3G would furnish 8-oxo-purines (8-oxo-4A and 8-oxo-4G; Fig. 1, dashed arrow)[22,29] rather than the desired purine nucleosides ara-4A and ara-4G. Therefore, it is of note that purine precursors 3A[22], 3I and 3G are highly resistant to alkaline hydrolysis, even in extremely alkaline (pH >12) solutions. Accordingly, we viewed this subtle, yet pronounced, difference in reactivity as an ideal source for chemical differentiation that could be exploited while building the canonical nucleobases on a preformed furanosyl-sugar scaffold. We envisaged sulfur—a critical element in the development of divergent ribonucleoside syntheses[22], with widespread use in prebiotic chemistry[22,30–35]—would hold the key to site-selective purine reduction.

Here, we demonstrate a divergent route to synthesise a complete set of canonical (A, G, C and U) nucleosides from one plausibly prebiotic reaction sequence. Interestingly, photochemical reduction of (intermediate) 8-mercaptopurines is observed to be highly efficient for the desired canonical purines (A and G), but not the non-canonical purine inosine (I). The facile prebiotic synthesis of ANA indicates that it may have played an important role during the origins of life, and the selective photochemical reduction of 8-mercaptopurines provides a physical mechanism for prebiotic nucleobase selection en route to the Watson–Crick base-pairing nucleosides.

## Results

**8-Mercaptopurine synthesis.** We suspected that addition of hydrogen sulfide ($H_2S$) to 3A, 3G and 3I in water would selectively introduce sulfur at the C8-carbon atom, and consequently direct regiospecific reduction of the canonical purine nucleobases on the preformed sugar scaffold (Fig. 1, blue arrows). Previously, Ikehara and Ogiso established that 3A reacts with liquid $H_2S$ in pyridine at 100 °C (sealed in a steel tube) to afford 8-mercapto-arabino-adenosine (ara-5A)[36]. However, 3A, 3G and 3I were remarkably stable to alkaline hydrolysis (Supplementary Figs. 1 and 2). Therefore, we began our investigation by exploring the mild, plausibly prebiotic, aqueous thiolysis of the prebiotic purine

**Fig. 1** Divergent prebiotic nucleotides synthesis. Red arrows: Previous work; a prebiotic pathway to cytidine-2′,3′-cyclic phosphate (1C), uridine-2′,3′-cyclic phosphate (1U), 8-oxo-adenosine-2′,3′-cyclic phosphate (2A) and 8-oxo-inosine-2′,3′-cyclic phosphate (2I)[16,22]. Dashed arrow: Hydrolysis of 8,2′-anhydropurines (3A, 3I and 3G) is not observed, which provides chemical differentiation from 2,2′-anhydropyrimidine (3C) that readily hydrolyses to β-arabino-adenosine (ara-4C). Blue arrows: This work; a prebiotic pathway to β-arabino-cytidine (ara-4C), β-arabino-uridine (ara-4U), β-arabino-adenosine (ara-4A), β-arabino-inosine (ara-4I) and β-arabino-guanosine (ara-4G)

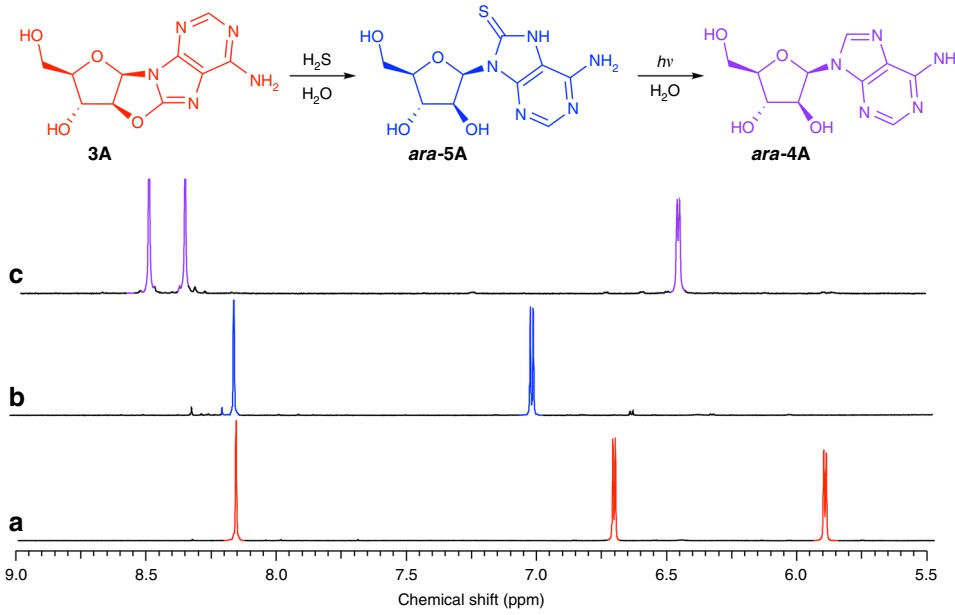

**Fig. 2** Prebiotic synthesis of *arabino*-adenosine (***ara*-4A**). $^1$H NMR spectra (600 MHz, 9:1 $H_2O/D_2O$, 25 °C, $\delta = 5.5–9.0$ ppm) showing: **a** anhydroadenosine (**3A**; red). **b** Crude 8-mercapto-*arabino*-adenosine (***ara*-5A**; blue) observed upon incubation of anhydroadenosine (**3A**; 35.7 mM) and $H_2S$ (670 mM, 60 °C, 7 d, pH 7). **c** Crude *arabino*-adenosine (***ara*-4A**; purple) after irradiation ($\lambda = 300$ nm) of 8-mercapto-*arabino*-adenosine (***ara*-5A**; 2 mM, room temperature, 16 h, pH 7)

| Table 1 Percentage conversions for nucleoside reactions | | | | | |
|---|---|---|---|---|---|
| **Conversion** | **H₂S** | **Conversion** | **254** | **300** | **H₂O₂** |
| **3A**→***ara*-5A** | 73% | ***ara*-5A**→***ara*-4A** | 70% | 66% | 85% |
| — | — | **β-*ribo*-5A**→**β-*ribo*-4A** | — | 70% | — |
| **3G**→***ara*-5G** | 83% | **3G→**→***ara*-4G** | — | 59%$^a$ | 89% |
| — | — | **β-*ribo*-5G**→**β-*ribo*-4G** | — | 80% | — |
| **3I**→***ara*-5I** | 78% | ***ara*-5I**→***ara*-4I** | — | 15% | 90% |
| — | — | **β-*ribo*-5I**→**β-*ribo*-4I** | — | 10% | — |
| **3C**→***ara*-4C** | 91%$^b$ | — | — | — | — |
| **α-*ribo*-3C**→**α-*ribo*-4C** | 52%$^c$ | — | — | — | — |
| **3C**→***ara*-6C** | 31%$^d$ | ***ara*-6C**→**3C** | — | — | Quant. |
| **α-*ribo*-3C**→**α-*ribo*-6C** | 84%$^{d,35}$ | **α-*ribo*-6C**→**α-*ribo*-3C** | — | — | Quant. |
| — | — | **β-*ribo*-6C**→**β-*ribo*-4C** | — | — | Quant. |
| — | — | **β-*ribo*-6C**→**β-*ribo*-9** | — | — | 76%$^e$ |
| ***ara*-4C**→***ara*-7U** | 40%$^f$ | ***ara*-7U**→***ara*-4U** | 8% | 4% | 78% |
| **α-*ribo*-4C**→**α-*ribo*-7U** | 63%$^f$ | **α-*ribo*-7U**→**α-*ribo*-4U** | — | — | 93% |
| **β-*ribo*-4C**→**β-*ribo*-7U** | 16%$^f$ | **β-*ribo*-7U**→**β-*ribo*-4U** | — | — | 78% |

Conversions were directly determined by $^1$H NMR (600 MHz) spectroscopy in the crude product mixture. Conversion observed upon: H₂S: reaction with H₂S (20 equiv.) in water (pH 7, 60 °C, 7 d); 254: irradiation at $\lambda = 254$ nm in water (38 °C, pH 6.5, 16 h); 300: irradiation at $\lambda = 300$ nm in water (38 °C, pH 6.5, 16 h); H₂O₂: reaction with H₂O₂ (3 equiv.) in water (pH 7, room temperature)
$^a$22 h irradiation
$^b$Conversion at room temperature. Conversion of ***ara*-4C** to ***ara*-7U** occurs at 60 °C over 7 d to afford a mixture of ***ara*-4C/*ara*-7U** (1.5:1) in 95% combined conversion
$^c$Unoptimised hydrolysis observed after 7 d incubation with H₂S (20 equiv.) in water (60 °C, pH 7), observed alongside 39% hydrolysis of **α-*ribo*-4C** to **α-*ribo*-7U** (91% combined **α-*ribo*-4C + α-*ribo*-7U**)
$^d$Conversion in formamide with H₂S (4 equiv.)
$^e$Buffered at pH 3 with glycine; **β-*ribo*-9** observed alongside partial hydrolysis to **β-*ribo*-4C** (24**%**)
$^f$Unoptimised thiolysis observed after 7 d incubation with H₂S (20 equiv.) at 60 °C and pH 7 to investigate comparative rates of cytidine C4-thiolysis. Residual starting material ***ara*-4C** (52%), **α-*ribo*-4C** (37%) and **β-*ribo*-4C** (84%), respectively, accounted (>90%) for the residual mass balance

precursor **3A**[22,33,35]. Incubation of H₂S (670 mM) with **3A** (35.7 mM) in water (pH 7, 60 °C, 7 d) afforded remarkably clean conversion of **3A** to ***ara*-5A** (73%; Fig. 2, Table 1 and Supplementary Fig. 3).

Intrigued by the remarkably clean, high-conversion thiolysis of **3A**, we next investigated the thiolysis (pH 7, 60 °C, 7 d) of **3G** and **3I**. We observed ***ara*-5G** (83%) and ***ara*-5I** (78%; 71% isolated yield after 5 d) (Table 1 and Supplementary Figs. 4 and 5). Next, 1:1 **3A/3I** was subjected to thiolysis and gave a high conversion to both ***ara*-5A** (66%) and ***ara*-5I** (75%) (Supplementary Fig. 6).

Even under these mild, aqueous conditions, highly efficient thiolysis of **3A**, **3G** and **3I** was observed, demonstrating that sulfur can be regioselectively introduced to the C8-carbon atom of purines under plausibly prebiotic conditions.

**Photochemical purine reduction**. For purine reduction, we initially investigated the effect of UV light on 8-mercaptopurines **5**, because the atmosphere of the early Earth lacked an ozone layer[37], allowing UV light ($\lambda > 204$ nm) to irradiate Earth's surface[38,39]. We envisaged that UV irradiation of ***ara*-5A** would lead

to π–π* excitation, followed by C–S bond fragmentation to afford an N-heterocyclic carbene tautomer of **ara-4A**. Pleasingly, when **ara-5A** (2.00 mM, pH 7) was irradiated ($\lambda = 300$ or $254$ nm) in water we observed extremely clean conversion to **ara-4A** in 66% and 70%, respectively, after 16 h (Fig. 2 and Supplementary Figs. 7–9). Notably, **ara-4A** was the only nucleoside product observed after irradiation. Irradiation ($\lambda = 300$ nm) of the inosine analogue **ara-5I** gave *arabino*-inosine **ara-4I** (Supplementary Fig. 10). However, upon complete consumption of **ara-5I**, the observed conversion to **ara-4I** (15%) was significantly lower than for **ara-4A** (66%) (Table 1). Next, the two-step thiolysis/irradiation sequence was investigated for **3G**. Anhydronucleoside **3G** (5.85 mM) was thiolysed (H$_2$S, pH 7, 60 °C, 7 d) and then irradiated ($\lambda = 300$ nm). The reaction was monitored until complete consumption of the intermediate (**ara-5G**) was observed (Supplementary Fig. 11). Once again, we observed a good conversion (59% after 22 h; Table 1) to the desired product *arabino*-guanosine (**ara-4G**). Irradiation of 1:1 **ara-5A**/**ara-5I** confirmed the disparity between the conversions observed for the reduction to yield the canonical nucleobases A and G and wobble base-pairing I: 62% and 13% conversion to **ara-4A** and **ara-4I**, respectively, was observed (Supplementary Fig. 12). Notably, the irradiation ($\lambda = 300$ or $254$ nm) of 1:1 **ara-4A**/**ara-4I** (Supplementary Fig. 13) demonstrated equal product stability with respect to UV irradiation (e.g., 73% each after 16 h at $\lambda = 300$ nm), and therefore did not account for the intriguing differential percentage conversion observed between the canonical (A and G) and non-canonical (I) 8-mercapto-nucleosides. Subsequently, we irradiated the *ribo*-mercaptopurines **ribo-5A**, **ribo-5G** and **ribo-5I**, and observed highly efficient photo-reduction of the A and G *ribo*-nucleosides, but poor yielding reduction of the I *ribo*-nucleoside (Table 1 and Supplementary Figs. 14–16).

**Quantum chemical studies**. The observed difference in the photo-reduction of **5A** and **5G** compared to **5I** cannot be directly connected to any remarkable differences in their UV-absorption features (Supplementary Figs. 117–119). However, quantum chemical studies [ADC(2)][40,41] and femtosecond transient absorption spectroscopy (FTAS) suggest that after initial UV excitation these purines undergo different singlet-to-triplet decay pathways.

UV excitation of mercaptopurines (**5**) was calculated to result in the population of low-lying excited singlet states ($\pi\pi_{CS}^*$, $\pi\pi_{RING}^*$ and $n\pi_{CS}^*$), which all exhibit significant spin–orbit coupling (SOC) with the triplet manifold (Supplementary Table 2 and Supplementary Discussion). Therefore, these singlet states are expected to undergo efficient intersystem crossing (ISC) to populate the triplet excited states of **5**, as noted for other thionucleosides and thionucleobases[42–45]. Two important triplet minima were identified in our calculations (Fig. 3). First, the $T_1(^3\pi\pi_{CS}^*)$ triplet minimum (Fig. 3a), which leads to C8-thiocarbonyl elongation ($\approx 0.1$ Å) and localisation of unpaired electrons on the $sp^3$-hybridised C8-carbon and sulfur atoms. Second, the $T_1(^3\pi\pi_{RING}^*)$ triplet minimum (Fig. 3a), which leads to pyrimidine ring puckering and localisation of unpaired electrons on the partially $sp^3$-hybridised C2- and C5-carbon atoms. The **ara-5A** $^3\pi\pi_{ring}^*$ triplet is significantly higher in energy than the **ara-5A** $^3\pi\pi_{CS}^*$ triplet; consequently, population of the $^3\pi\pi_{CS}^*$ triplet is predicted to be the dominant pathway during ISC following UV excitation of **ara-5A** (Fig. 3d). Accordingly, we expect that C8-photo-reduction is triggered in the $^3\pi\pi_{CS}^*$ triplet state of 8-mercaptopurines **5**. In contrast to **ara-5A**, which is expected to only populate the $^3\pi\pi_{CS}^*$ triplet state, **ara-5I** and **ara-5G** are predicted to populate both triplet states ($^3\pi\pi_{CS}^*$ and $^3\pi\pi_{ring}^*$). Optimisation of the mercaptopurine excited state

geometries indicated that the $T_1$ hypersurface of **ara-5I** (Fig. 3c) and **ara-5G** (Supplementary Fig. 51) are similar in topography to those reported for thiopyrimidine nucleobases[46]. Consequently, we expected **ara-5I** and **ara-5G** to exhibit similar UV excitation behaviour to thiopyrimidines, which are observed to undergo two competing excited state decay pathways[46].

Short excited state lifetimes are expected for the redox active $T_1(^3\pi\pi_{CS}^*)$ triplet states, due to their very large SOC values (**ara-5A**: 99.5 cm$^{-1}$; **ara-5I**: 81.4 cm$^{-1}$) and low (0.23 eV) state-crossing barrier to the electronic ground state ($S_0$). Conversely, the **ara-5I** $T_1(^3\pi\pi_{RING}^*)$ triplet state exhibits a very low SOC value (1.44 cm$^{-1}$) and is predicted to be much longer lived. Moreover, the **ara-5I** $T_1(^3\pi\pi_{RING}^*)$ triplet minimum is estimated to be 0.2 eV lower in energy than the redox active **ara-5I** $T_1(^3\pi\pi_{CS}^*)$ triplet minimum, and predominant population of the non-redox active $T_1(^3\pi\pi_{RING}^*)$ triplet minimum is expected following UV excitation of **ara-5I** because these two states are only separated by a moderate (0.55 eV) energy barrier.

Upon first inspection it might seem surprising that, like **ara-5I**, both triplet states can be populated in **ara-5G** given its observed efficient photo-reduction. However, the **ara-5G** $T_1(^3\pi\pi_{RING}^*)$ and $T_1(^3\pi\pi_{CS}^*)$ triplet minima are calculated to be nearly isoenergetic, and therefore these minima are expected to easily interconvert. Accordingly, efficient photo-reduction of **ara-5G** is thought to occur by continual repopulation of the redox active **ara-5G** $T_1(^3\pi\pi_{CS}^*)$ triplet minimum.

**Femtosecond transient absorption spectroscopy**. We next sought to verify our theoretical predictions by FTAS for **ara-5I** (Fig. 3e) and **ara-5A** (Fig. 3f). The absorbance ($\lambda = 380$–$450$ nm) observed in the first picosecond following **ara-5A** excitation matches the position and structure of the excited state absorption (ESA) spectrum simulated from the $S_1(^1\pi\pi_{CS}^*)$ singlet minimum of **ara-5A** (Fig. 3f, inset). Initial singlet ($^1\pi\pi_{CS}^*$) state population following photo-excitation is also consistent with our calculated vertical excitation energies (Supplementary Table 1), which indicate that $\lambda = 290$–$300$ nm excitation would primarily populate this state (Fig. 3e). At probe time delays >2 ps after incident excitation of **ara-5A**, the absorbance evolved to cover a broader range of probe wavelengths, and the FTAS spectrum exhibits two characteristic features near $\lambda = 360$ and $480$ nm. Although these features are partly covered by the bleaching bands, they match the simulated **ara-5A** $T_1(^3\pi\pi_{CS}^*)$ triplet state ESA spectrum (Fig. 3f, inset), which indicates population of this triplet state. The observed excited state lifetime of **ara-5A** ($\approx 70$ ns at $\lambda = 510$ nm) is also consistent with the predicted efficient photo-relaxation of this triplet state, which is characterised by high SOC with the electronic ground state.

Although excitation at $\lambda = 290$ nm could potentially populate both **ara-5I** singlet states (i.e., $^1\pi\pi_{CS}^*$ and $^1\pi\pi_{ring}^*$), we did not observe any features in the FTAS spectrum that would correspond to the $^1\pi\pi_{CS}^*$ singlet state. No significant change in absorbance was recorded during the first 0.4 ps (Fig. 3e). We were unable to simulate the ESA spectrum for the **ara-5I** $S_1(^1\pi\pi_{ring}^*)$ singlet state (its minimum-energy geometry coincided with the $S_1/S_0$ conical intersection), but we anticipate that population of this singlet state dominates the 0.4 ps immediately after **ara-5I** excitation. The emergence of a sharp band at $\lambda = 350$ nm between 0.6 and 1.0 ps was assigned to population of the redox active $T_1(^3\pi\pi_{CS}^*)$ triplet state, which could be efficiently accessed from the initial $S_1(^1\pi\pi_{ring}^*)$ singlet state because of the molecular orbital change associated with an $S_1 \rightarrow T_1$ transition[47]. The redox active **ara-5I** $T_1(^3\pi\pi_{CS}^*)$ triplet minimum is observed to be efficiently depopulated over the next 6 ps. The excited state population transfers to the lower-energy **ara-5I** $T_1(^3\pi\pi_{ring}^*)$

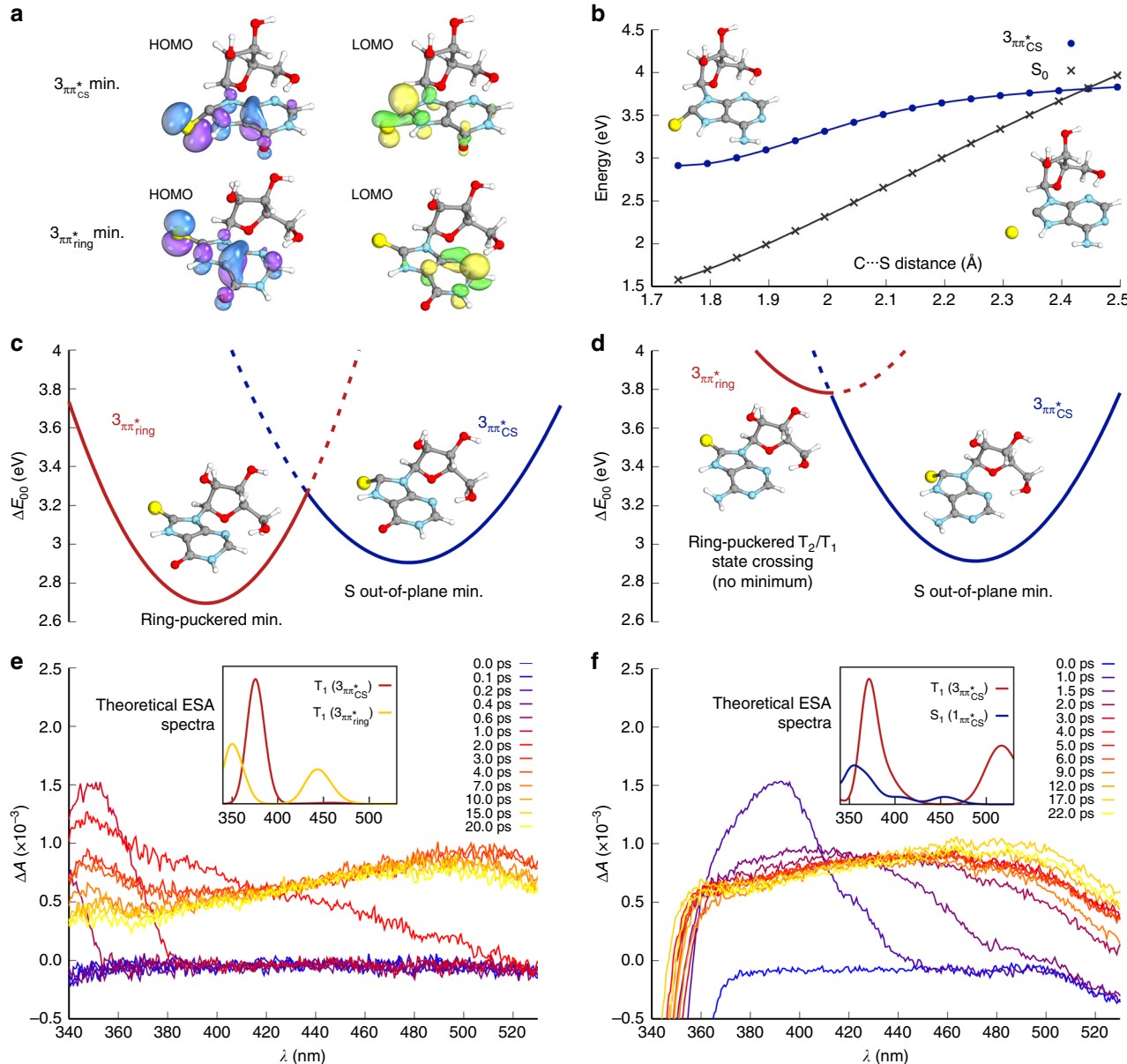

**Fig. 3** Photochemical properties of **ara-5I** and **ara-5A**. **a** Molecular orbitals for $T_1$ states ($^3\pi\pi_{CS}^*$ and $^3\pi\pi_{ring}^*$) of **ara-5I**. **b** Calculated potential energy profile of **ara-5A** $T_1$ $^3\pi\pi_{CS}^*$ state C–S bond homolysis furnishing triplet sulfur atom and singlet C8-carbene tautomer of **ara-4A**. Blue line = $T_1$-state energy, black line = ground-state energy. The minimum-energy path along the C–S distance (Å) was obtained by calculation at the ADC(2)/cc-pVTZ level of theory. **c** Calculated **ara-5I** triplet excited state $T_1$ topography. Parabolas fitted to calculated $T_1$ minima energies and optimised $T_2/T_1$ minimum-energy state crossing, which represents the transition state between these minima. **d** Calculated **ara-5A** triplet excited state $T_1$ topography. No ring-puckered minimum was found on the $T_1$ hypersurface due to the higher $^3\pi\pi_{ring}^*$ state energy. **e–f** FTAS recorded between $\lambda = 340$ and 530 nm with $\lambda = 290$ nm excitation (pump) pulses. **e** FTAS of **ara-5I** recorded between $\lambda = 340$ and 530 nm with $\lambda = 290$ nm excitation (pump) pulses. Inset: Calculated excited state absorption spectra for the two possible configurations of **ara-5I**. **f** FTAS of **ara-5A** recorded between $\lambda = 340$ and 530 nm with $\lambda = 290$ nm excitation (pump) pulses. Inset: Calculated excited state absorption spectra for the singlet and triplet states of **ara-5A**

triplet minimum (Fig. 3e), which is confirmed by the long excited state lifetime ($\approx$12 µs at $\lambda = 440$ nm) and the two characteristic bands ($\lambda_{max} = 350$ and 490 nm) in the **ara-5I** FTAS measurements, which are both consistent with the simulated ESA spectrum for the redox inactive triplet state (Fig. 3e, inset). The estimated excited state lifetime associated with the $T_1(^3\pi\pi_{ring}^*)$ triplet state is thought to be sufficient to enable bimolecular reactions and, consequently, the photochemical degradation pathways in **ara-5I**, which are not observed for **ara-5A** or **ara-5G**. Photo-reduction appears to correlate directly with population of the $T_1(^3\pi\pi_{CS}^*)$ triplet state for **ara-5A**, and we anticipate that

the same triplet state would be responsible for photo-reduction in **ara-5I** and **ara-5G**. In particular, the observed short-lived population of the **ara-5I** $T_1(^3\pi\pi_{CS}^*)$ triplet state (0.6–7.0 ps) likely explains the poor photo-reduction observed for this nucleoside.

It appears probable that C–S bond homolysis in **ara-5A**, initiated by population of the $T_1(^3\pi\pi_{CS}^*)$ triplet minimum, would furnish the singlet C8-carbene tautomer of purine **ara-4A** (Fig. 3b). Our calculations indicate that C–S bond homolysis requires 0.9 eV to liberate the triplet sulfur atom. Excitation at $\lambda = 300$ nm would excite **ara-5A** ~1.2 eV above the $T_1(^3\pi\pi_{CS}^*)$

**Fig. 4** Thiolysis and oxidation of pyrimidine nucleosides. **a** i. H$_2$S thiolysis of *arabino*-anhydrocytidine (**3C**) in water (pH 7) furnished β-*arabino*-cytidine (***ara*-4C**; quant.) rapidly; ii. ***ara*-4C** then undergoes slow addition of H$_2$S to afford 4-thio-β-*arabino*-uridine (***ara*-7U**). iii. H$_2$O$_2$ oxidation of ***ara*-7U** in water (pH 7) furnished *arabino*-uridine (***ara*-4U**, 78%). iv. Thiolysis of **3C** in formamide yielded 2-thio-β-*arabino*-cytosine (***ara*-6C**, 31%). v. H$_2$O$_2$ oxidation of ***ara*-6C** in water (pH 7) furnished **3C** (quant.). **b** i–ii. H$_2$S thiolysis of *ribo*-anhydrocytidine (α-*ribo*-**3C**) in water (pH 7) furnished α-*ribo*-cytidine (α-*ribo*-**4C**), which then undergoes addition of H$_2$S to afford 4-thio-α-*ribo*-uridine (α-*ribo*-**7U**) in 91% combined conversion. iii. H$_2$O$_2$ oxidation of α-*ribo*-**7U** in water (pH 7) furnished α-*ribo*-uridine (α-*ribo*-**4U**, 93%). iv–v. Sutherland and co-workers reported the synthesis of 2-thio-β-*ribo*-cytidine (β-*ribo*-**6C**) by thiolysis of *ribo*-anhydrocytidine (α-*ribo*-**3C**) in formamide, followed by aqueous irradiation (λ = 254 nm)[35]. vi. H$_2$O$_2$ oxidation of 2-thio-β-*ribo*-cytosine (β-*ribo*-**6C**) furnished canonical ribonucleoside β-cytidine (β-*ribo*-**4C**, quant.) between pH 7 and 9. vii. Conversely, H$_2$O$_2$ oxidation of β-*ribo*-**6C** predominantly furnished pyrimidime 4-amino-pyrimidine-riboside (β-*ribo*-**9**, 76%) at pH 3. viii. Due to the proximity of the nucleobase and C2′-hydroxyl moieties, H$_2$O$_2$ oxidation of α-*ribo*-**6C** furnished α-*ribo*-**3C** (quant.). ix–x. The sequential reaction of H$_2$S and H$_2$O$_2$ with β-*ribo*-**4C** in water (pH 7) furnished β-*ribo*-**4U**

triplet minimum, which would be sufficient to promote efficient photo-reduction. In principle, C–S bond homolysis (Fig. 3b) would be reversible, however tautomerisation of the intermediate C8-carbene to ***ara*-4** would prevent reformation of ***ara*-5**.

**8-Mercaptopurine oxidation.** Encouraged by the photochemical reduction of ***ara*-5A**, ***ara*-5G** and ***ara*-5I**, we investigated how to increase the efficiency of purine reduction. Simplicity and pre-biotic plausibility led us to consider hydrogen peroxide (H$_2$O$_2$) in nucleobase reduction[48–50]. In our hands, the reported reaction of H$_2$O$_2$ and ***ara*-5A** in acidic methanol[36] gave ***ara*-4A** (50% after 16 h, Supplementary Fig. 17). Pleasingly, oxidative dis-proportionation of ***ara*-5A** (50 mM) and ***ara*-5I** (50 mM) under plausibly prebiotic conditions (H$_2$O$_2$ (3 equiv), pH 7, water, room temperature, 3 h) afforded excellent conversion to ***ara*-4A** (85%) and ***ara*-4I** (90%), respectively (Table 1 and Supplementary Figs. 18 and 19). Reaction of 1:1 ***ara*-5A**/***ara*-5I** afforded ***ara*-4A**

(88%) and ***ara*-4I** (91%) (Supplementary Fig. 20). Next, we investigated the one-pot thiolysis/oxidation of **3G**. Incubation of **3G** (35 mM) with H$_2$S (714 mM) in water (pH 7, 60 °C, 7 d) then addition of H$_2$O$_2$ (375 μmol; pH 7, room temperature, 2 h) furnished ***ara*-4G** (89%) (Table 1 and Supplementary Fig. 21).

**Pyrimidine thiolysis.** Having demonstrated the efficient con-version of **3A**, **3G** and **3I** to *arabino*-purines ***ara*-4A**, ***ara*-4G** and ***ara*-4I**, respectively, we next investigated the reactivity of pyr-imidine precursor **3C** under comparable conditions. Sutherland and co-workers reported that the reaction of α-*ribo*-anhy-drocytidine (α-*ribo*-**3C**) with sodium hydrogen sulfide (NaHS) in formamide at 50 °C yields 2-thio-α-cytidine (α-*ribo*-**6C**; 84%; Fig. 4b.iv)[35]. Similarly, upon submitting **3C** to these conditions (NaHS, formamide, 50 °C), we observed conversion to ***ara*-4C** (24%) and 2-thio-*arabino*-cytidine (***ara*-6C**; 31%) (Fig. 4a.iv; Supplementary Fig. 24). Conversely, incubation of **3C** (28 mM)

with $H_2S$ (0.14 M) in water (pH 7, 60 °C) yielded mostly **ara-4C** (68%) alongside some conversion of **ara-4C** to 4-thio-*arabino*-uridine (**ara-7U**; 22%) after 2 d (Fig. 4a.i–ii; Supplementary Fig. 25); <5% **ara-6C** was observed. Further conversion of **ara-4C** to **ara-7U** was observed upon prolonged incubation (57:38 **ara-4C**/**ara-7U** after 7 d; Fig. 4a.ii). Although in formamide the addition of NaHS to **3C** furnishes **ara-6C**, in water rapid hydrolysis of **3C** to **ara-4C** occurs, which subsequently undergoes slow nucleophilic substitution at the C4-carbon atom to furnish **ara-7U**.

The observed C4-thiolysis of **ara-4C** to **ara-7U** is not limited to the β-*arabino*-stereochemistry shown in Fig. 4a.ii. Submitting α-*ribo*-**3C** (28 mM) to the same aqueous thiolysis conditions afforded α-*ribo*-cytidine (α-*ribo*-**4C**, 52%; Fig. 4b.ii) and 4-thio-α-*ribo*-uridine (α-*ribo*-**7U**, 39%; Fig. 4b.ix). Incubation of **ara-4C**, β-*ribo*-**4C** and α-*ribo*-**4C** with $H_2S$ (pH 7) led to partial (unoptimised) conversion to the 4-thiouridines **ara-7U** (7 d, 40%), β-*ribo*-**7U** (7 d, 16%) and α-*ribo*-**7U** (8 d, 63%), respectively (Table 1), alongside recovered cytidine starting material [**ara-4C** (52%), β-*ribo*-**4C** (84%) and α-*ribo*-**4C** (37%)] (Supplementary Figs. 28–30). We were surprised to observe slower C4-thiolysis for the canonical *trans*-1′,2′-isomer (β-*ribo*-**4C**) than for the *cis*-1′,2′-isomers (α-*ribo*-**4C** and **ara-4C**). Simultaneous thiolysis of 1:1 **ara-4C**/β-*ribo*-**4C** verified this rate difference; after 7 d more **ara-7U** (52%) than β-*ribo*-**7U** (18%) was observed (Supplementary Fig. 31), indicating the relative nucleobase/C2′-OH orientation affects the rate of nucleophilic substitution at the distal C4-position of cytidine nucleotides. Finally, we incubated pyrimidine **3C** and purine **3A** (1:1) with $H_2S$ (pH 7, 60 °C, 7 d). Gratifyingly, **ara-4C** (70%), **ara-7U** (30%) and **ara-5A** (71%) were cleanly furnished (Supplementary Fig. 27).

**Pyrimidine oxidation.** An efficient protocol to convert **ara-7U** to **ara-4U** would indicate that nucleosides **ara-4C** and **ara-4U** could be readily generated in comparable yields from thiolysis of **3C** alongside conversion of **3A**/**3G** to **ara-4A**/**ara-4G**, respectively. We envisioned that the thiocarbonyl moiety of **ara-7U** would be readily oxidised, however we suspected that oxidation would activate the pyrimidine nucleobase to hydrolysis (rather than reduction as observed for the purines). To test our hypothesis, we incubated **ara-7U** (50 mM), β-*ribo*-**7U** (50 mM) and α-*ribo*-**7U** (50 mM) with $H_2O_2$ (0.15 M) in water (pH 7, room temperature, 3.5 h). Each reaction proceeded smoothly to give the respective uridine (**ara-4U** (78%); β-*ribo*-**4U** (78%); α-*ribo*-**4U** (93%); Table 1, Fig. 4 and Supplementary Figs. 38, 40–41), which validated our prediction. We next investigated thiolysis and selective hydrolysis in a one-pot reaction: **3C** (35.7 mM) was incubated with $H_2S$ (0.71 M) in water (pH 7, 60 °C, 7 d) and then $H_2O_2$ (375 µmol) was added. We observed concomitant formation of **ara-4C** (62%) and **ara-4U** (25%) (Supplementary Fig. 39).

The reactions of 2-thiocytidines **ara-6C** and α-*ribo*-**6C** with $H_2O_2$ (150 mM) in water (pH 7, room temperature, 2 h) cleanly regenerated anhydrocytidines **3C** (82%; Fig. 4a.v) and α-*ribo*-**3C** (80%; Fig. 4b.viii), respectively, demonstrating a clear switch in reactivity relative to **ara-7U** due to the proximity of the C2′-hydroxyl and thiocarbonyl moieties. Therefore, it is of note that canonical cytidine β-*ribo*-**6C** has an *anti*-1′,2′-disposition, and we expected to observe hydrolysis during $H_2O_2$ oxidation of β-*ribo*-**6C**. As expected, incubation of β-*ribo*-**6C** (38.4 mM) with $H_2O_2$ (232 mM) in phosphate buffer (pH 7–9, room temperature, 7 h) afforded a quantitative conversion to β-*ribo*-**4C** (Fig. 4b.vi; Supplementary Fig. 44). Interestingly, buffering the oxidation at pH 3 with glycine afforded β-*ribo*-**4C** (24%) alongside 4-amino-pyrimidine-riboside (β-*ribo*-**9**; 76%;

Fig. 4b.vii; Supplementary Fig. 44). It is likely that cytidine protonation switches on C2-reduction by promoting access to the carbene intermediate required for reduction (Fig. 4b.vii)[51]. This hypothesis is supported by the simultaneous oxidation of **ara-7U** and β-*ribo*-**6C** with $H_2O_2$ (pH 3, glycine buffer, room temperature), which affords β-*ribo*-**4C** (20%) and β-*ribo*-**9** (72%) from β-*ribo*-**6C**, but exclusively **ara-4U** (92%) from **ara-7U** (Supplementary Fig. 45).

It is particularly interesting, with respect to the origins of life, that purine C8-reduction is facile and quantitative at neutral pH, whereas pyrimidine C2-reduction is only observed at low pH. Consequently, the reaction of β-*ribo*-**6C** with $H_2O_2$ at neutral pH furnishes canonical β-*ribo*-**4C** cleanly, but protects against, and even reverses, the formation of non-canonical **ara-6C** and α-*ribo*-**6C**. This provides a facile and selective method to convert β-*ribo*-**6C**[35] to canonical nucleoside β-*ribo*-**4C**.

**Concomitant purine and pyrimidine synthesis.** To form information-rich ANA nucleic acid oligomers, all four Watson–Crick base-pairing nucleosides (**ara-4A**, **ara-4C**, **ara-4G** and **ara-4U**) need to accrue at the same time in the same environment, ideally from the same set of chemical reactions. We reacted 1:1:1 **3C**, **3A** and **3G** with $H_2S$ (pH 7, 60 °C; Supplementary Fig. 46), and after 7 d we observed **ara-5A** (65%), **ara-5G** (62%), **ara-4C** (55%) and **ara-7U** (35%). Subsequent incubation with $H_2O_2$ (pH 7, room temperature, 2 h) gave the desired Watson–Crick base-pairing products **ara-4A** (53%), **ara-4G** (62%), **ara-4C** (47%) and **ara-4U** (35%) without purification or isolation of intermediate products (Fig. 5). Thus, we achieved a plausibly prebiotic divergent synthesis of Watson–Crick base-pairing nucleosides, and have marked ANA as a likely candidate for the nucleic acid of early evolution.

**Discussion**
Through the reaction of pyrimidine and purine precursors (**3C** and **3A**, respectively), which can be accessed divergently from a single prebiotic substrate[16,22], we have elucidated an efficient plausibly prebiotic method to simultaneously form *arabino*-pyrimidine and *arabino*-purine nucleosides. Both the purine and pyrimidine nucleosides are delivered with regiospecific glycosylation on a furanose-specific arabinose sugar moiety. The reaction of $H_2S$ in water unlocks a purine-specific C8-reduction, by either UV irradiation or $H_2O_2$ oxidation; $H_2O_2$ oxidation is especially high yielding, however, UV irradiation caused selective destruction of **ara-5I**/β-*ribo*-**5I** over **ara-5A**/β-*ribo*-**5A** and **ara-5G**/β-*ribo*-**5G**. The remarkable purity of the *arabino*-purines delivered by either route is highly encouraging, but the direct mechanism for canonical purine nucleobases selection by UV light provides a mechanism (based in the physical properties of the purine bases) for nucleobase selection prior to their incorporation into nucleic acid biopolymers at the origins of life.

In water $H_2S$ also undergoes selective, but slow, addition to the cytidine C4-position, and—when coupled with oxidative hydrolysis—this provides a selective mechanism to convert cytidines to uridines. Importantly, $H_2S$ did not convert A/G to I/xanthine (X) demonstrating another remarkable, but essential, reactivity difference between the canonical nucleosides.

Although our focus has been on developing new chemistry, not on assessing geochemical boundary conditions, we note that the availability of $H_2S$ and $H_2O_2$ are both plausibly prebiotic, and although their sequential reaction implicates different redox states this is not geochemically implausible. The entire planet is at redox disequilibrium, and layered redox gradients are geochemically common. Outgassing volatile compounds (from Earth's interior) are likely to have played a critical role in determining prebiotic

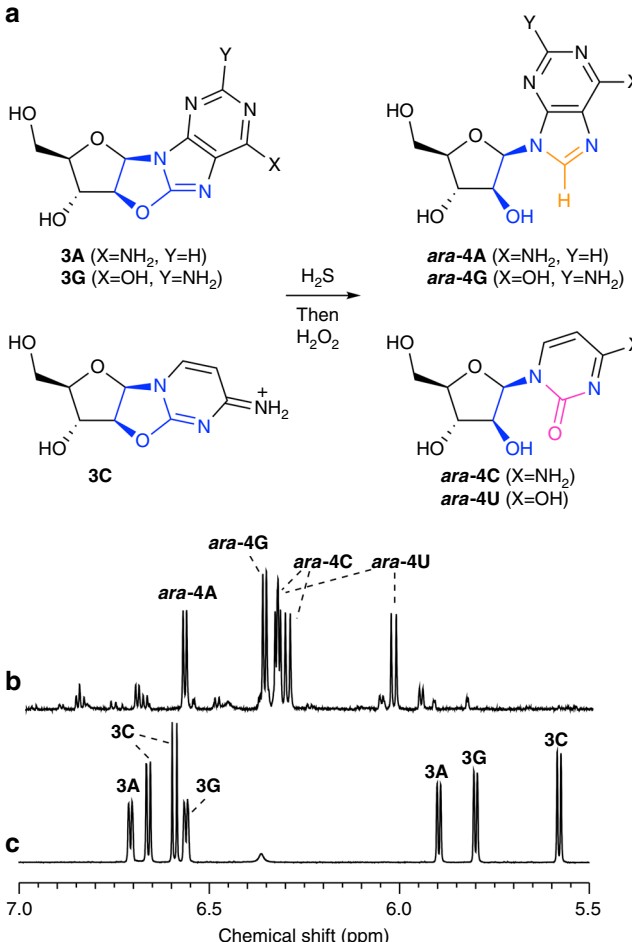

**Fig. 5** Concomitant synthesis of four Watson–Crick base-pairing nucleosides. **a** One-pot reaction of an equimolar mixture of anhydroadenosine (**3A**), anhydroguanosine (**3G**) and anhydrocytidine (**3C**) with $H_2S$ (20 equiv., pH 7) in water, then $H_2O_2$ (20 equiv., pH 7) in water affords adenosine **ara-4A** (53%), guanosine **ara-4G** (62%), cytidine **ara-4C** (47%) and uridine **ara-4U** (35%). The isourea moieties (blue) of the anhydropurines (**3A/G**) and anhydropyrimidine (**3C**) were differentiated in situ to afford the canonical Watson–Crick base-pairing nucleobase moieties regiospecifically on a furanosyl-sugar scaffold, by reduction (orange) and hydrolysis (magenta), respectively. **b** $^1H$ NMR spectrum (600 MHz, 9:1 $H_2O/D_2O$, 25 °C, $\delta = 5.5$–7.0 ppm) showing the prebiotic synthesis of cytidine **ara-4C**, uridine **ara-4U**, adenosine **ara-4A** and guanosine **ara-4G** following the sequential addition of $H_2S$ and $H_2O_2$ to anhydrocytidine **3C**, anhydroadenosine **3A** and anhydroguanosine **3G** in water. **c** $^1H$ NMR spectrum (600 MHz, 9:1 $H_2O/D_2O$, 25 °C, $\delta = 5.5$–7.0 ppm) of substrates anhydrocytidine **3C**, anhydroadenosine **3A** and anhydroguanosine **3G** in water

chemistry, and local redox environments (for example, due to volcanic outgassing, meteorite impacts, photo-oxidation and atmospheric water dissociation and hydrogen escape) are expected to provide significant variation from the global average[52]. $H_2S$ is produced through a distinct mechanism from $H_2O_2$, thus discrete redox zones could be readily established. The oxidation state of early Earth (Hadean) magmas are not well constrained, however, zircons that pre-date the known rock record suggest average oxygen fugacities may have been similar to the present-day conditions[53]. Accordingly, sulfide-leaching and $H_2S$-outgassing would have occurred on the early Earth, with sulfur outgassing rates as high as $10^{11.5}$ cm$^{-2}$ s$^{-1}$ possible during major volcanogenic emplacement of basaltic plains[54]. Equally, photo-

dissociation generates $H_2O_2$ from water, and $H_2O_2$ is a likely key environmental oxidant prior to global oxidation[37,50,55]. It seems reasonable to suppose that distinct (redox) reactivity could be controlled by geochemical localisation and the different physicochemical processes that yield and accumulate feedstock molecules. Geochemical $H_2S$ outgassing and atmospheric $H_2O_2$ production are well suited to this specific localisation. Moreover, it has recently been proposed that $H_2O_2$ can accumulate within ices in an anoxic atmosphere[56]. Melting $H_2O_2$-rich ices could augment $H_2O_2$ delivery into aqueous environments and, in principle, could help to provide a mechanism for the sequential delivery of $H_2S$ and $H_2O_2$ into, for example, a flowing stream system or pool. Although the reactions of $H_2S$ and $H_2O_2$ must occur in sequence to achieve purine reduction by the described oxidative mechanism, it is of note that the purine anhydronucleotides (**3A** and **3G**) were observed to be stable to $H_2O_2$ oxidation and the reduced purines (**ara-4A** and **ara-4G**) were observed to be stable to $H_2S$ addition. Therefore, cycling material between these redox conditions (or across this redox gradient) is not considered to be problematic for this chemistry; indeed, a one-pot two-step $H_2S/H_2O_2$ reduction has been demonstrated. Importantly, photochemical mercaptopurine reduction does not require two redox states, only $H_2S$ and UV light (at $\lambda = 250$–300 nm), both of which are expected to be in adequate (simultaneous) supply on the early Earth when sulfur outgassing rates are less than $10^{11.5}$ cm$^{-2}$ s$^{-1}$. The localised enhancement in $H_2S$ can be achieved in surface hydrothermal systems with shallow water reservoirs (to allow UV penetration)[54]. In terms of the UV-light dose environment, we note that in the laboratory only six of the available sixteen lamps in the Rayonet irradiation chamber were employed, which corresponds to 39,000 erg s$^{-1}$ cm$^{-2}$ and 30,900 erg s$^{-1}$ cm$^{-2}$ at $\lambda = 254$ nm and 300 nm, respectively. The integrated surface flux ($\lambda = 200$–300 nm) delivered to the Earth by the early Sun is estimated to be about 2700 erg s$^{-1}$ cm$^{-2}$ (and within roughly an order of magnitude of the experimental apparatus), which provides ample flux to achieve the desired transformations and preserve prebiotic plausibility[39]. Crucially, the two distinct mercaptopurine **5** reductions demonstrate, not only the specific value of sulfur in prebiotic nucleoside synthesis, but also, more generally, the value of chemical redundancy. Two mechanisms for mercaptopurine reduction that operate under different conditions but both furnish the same purine products renders the overall transformation more robust, which may be especially important to consider in the context of the origins of life. Chemical redundancy is highly likely to improve pathway or network robustness, which may be an essential feature of sustained protometabolism in a (geo)chemically fluctuating environment[3,18].

The different reactions observed between **3C** and **3A/G** with $H_2S$ is ideally suited to concomitant synthesis of the canonical nucleobases on preformed sugar scaffolds. Although the *arabino*-stereochemistry is not found in extant genetics, it is highly plausible that ANA could have been a precursor to RNA in early life or that early co-evolution of mixed RNA/ANA systems could have been superseded by RNA/DNA systems. The simplicity and efficiency of the synthesis of A, C, G and U arabinosides indicates that further investigations into the synthesis of **3G** and the potential for ANA and RNA co-evolution are both warranted.

## Methods

**General procedure A. Thiolysis.** Sodium hydrosulfide or disodium sulfide non-ahydrate (20 equiv.) was dissolved in $H_2O/D_2O$ (9:1) at pH 7. Nucleoside(s) (1 equiv.) was added and incubated at pH 7 and 60 °C. The reaction progress was monitored periodically by NMR spectroscopy. Upon completion, the reaction was cooled to room temperature and sparged of hydrogen sulfide with nitrogen or argon gas. The solution was adjusted to pH 6.5 and analysed by NMR spectroscopy.

**General procedure B. UV irradiation**. A degassed aqueous solution of nucleoside (s) (2.00 mM, pH 6.5) was irradiated in a Rayonet reactor (SNE Ultraviolet Co.) housing six RPR-2537A or RPR-3000A lamps (with principal emission at $\lambda = 254$ nm and $\lambda = 300$ nm, respectively) at 38 °C under an argon atmosphere. After irradiation, the reaction was allowed to cool to room temperature, and then lyophilised. The lyophilisate was dissolved in $D_2O$ (500 μL) and analysed by NMR spectroscopy. A solution of potassium hydrogen phthalate (0.100 M, 50.0 μL, 5.00 μmol in $D_2O$) was added as an internal NMR standard and NMR spectra were reacquired.

**General procedure C. Nucleoside oxidation**. Nucleoside(s) (1.0 equiv., 50 mM) and potassium hydrogen phthalate (0.2 equiv.) were dissolved in $H_2O/D_2O$ (9:1) or buffer solution. The pH of the solution was adjusted to pH 7 and NMR spectra were acquired. Hydrogen peroxide (30% w/w solution in $H_2O$) was added and NMR spectra were periodically acquired at pH 7.

**Computational methods**. The vertical excitation energies, excited state geometries and excited state harmonic vibrational frequencies of *ara*-5I, *ara*-5A and *ara*-5G were computed by using the algebraic diagrammatic construction to the second-order method [ADC(2)][40,41,57], and the cc-pVTZ basis set. The MP2/cc-pVTZ method was used to optimise the corresponding ground-state geometries. Spin–orbit coupling matrix elements were calculated at the CASPT2/SA-CASSCF (10,9)/cc-pVTZ-DK level[58], including the second-order Douglas–Kroll–Hess transformation to account for scalar relativistic effects. Calculated ESA spectra were obtained by convolution of vertical excitation energies and oscillator strengths with normalised Gaussian functions (0.20 eV half-width). Vertical excitation energies necessary for the ESA spectra were calculated from the corresponding excited state minima and 11 excitations were taken into account in each case. ESA cross-sections were generated using the GaussSum programme[59]. Minimum-energy crossing point (MECPs) geometries were optimised with the in-house implementation of the method proposed by Levine, Coe and Martinez[60]. The energies and gradients in electronically excited states were computed at the ADC(2) level, whereas the corresponding properties for electronic ground states were obtained at the MP2 level in the MECP geometry optimisations[41]. The MECP optimisation steps were performed with the Broyden–Fletcher–Goldfarb–Shanno quasi-Newton scheme available in the internal optimiser of Turbomole 7.1[61]. All ADC(2) and MP2 calculations were performed with Turbomole 7.1, and Molcas 8.0[62] was employed for all the CASPT2/SA-CASSCF calculations. Graphical representations of the molecular geometries and orbitals were generated with IboView[63].

**Femtosecond transient absorption spectroscopy**. FTAS experiments were performed with a Solstice Ace pulsed laser system (Spectra-Physics, Newport Co.), which produces 97 fs pulses at $\lambda = 800$ nm. To generate the white light continuum probe pulses ($\lambda = 320-700$ nm) in the Helios-Fire spectrometer (Ultrafast Systems, LLC), a fraction of the fundamental beam was focused on a thin $CaF_2$ crystal. The $\lambda = 290$ nm excitation (pump) pulses were generated by passing the remainder of the fundamental beam through an optical parametric amplifier (TOPAS, Light Conversion, Ltd.). FTAS experiments of aqueous samples of *ara*-5A (4 mM, pH 7.4) and *ara*-5I (4 mM, pH 7.4) were performed in 2 mm optical path length quartz cuvettes (Starna Cells, Inc.). LabView Surface Xplorer software (Ultrafast Systems, LLC) was used to reduce the FTAS data, apply corrections for group velocity dispersion of the white light probe, and extract and analyse spectra and their time dependence.

**X-ray diffraction**. All diffraction data were collected by using a four-circle Agilent SuperNova (Dual Source) single crystal X-ray diffractometer with a micro-focus $CuK_\alpha$ X-ray beam ($\lambda = 1.54184$ Å) and an Atlas CCD detector. The crystal temperature was controlled by using an Oxford Instruments Cryojet5. Unit cell determination, data reduction and analytical numeric absorption correction using a multifaceted crystal were carried out using the CrysAlisPro programme[64]. The crystal structures were solved with the ShelXS programme and refined by least squares on the basis of $F^2$ with the ShelXL programme[65]. All non-hydrogen atoms were refined anisotropically by the full-matrix least-squares method. Hydrogen atoms affiliated with oxygen and nitrogen atoms were refined isotropically in positions identified by the difference Fourier map, or in geometrically constrained positions. Hydrogen atoms associated with carbon atoms were refined isotropically in geometrically constrained positions.

## Data availability

The authors declare that data supporting the findings of this study are available within the paper and its Supplementary Information files and figures. X-ray crystallographic data were deposited at the Cambridge Crystallographic Data Centre (CCDC) under the following CCDC deposition numbers: 1586272 (*ara*-5A, Supplementary Fig. 120) and 1836032 (3′,3-anhydro-guanosine (12); the high pH isomerisation product of 3G, Supplementary Fig. 121). These can be obtained free of charge from CCDC via https://www.ccdc.cam.ac.uk/structures/.

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

## Acknowledgements

This work was supported in part by the Simons Foundation (318881 to M.W.P., 494188 to R.S. and 290360 to D.D.S.) and the Engineering and Physical Sciences Research Council (EP/K004980/1 to M.W.P.). The authors thank Dr. K. Karu for assistance with mass spectrometry and Dr. A.E. Aliev for assistance with NMR spectroscopy. Z.R.T. and D.D.S. would like to thank D. Bucher, and to acknowledge the Harvard Origins of Life Initiative.

## Author contributions

M.W.P. conceived the research. M.W.P. and S.J.R. designed and analysed the experiments. S.J.R. and S.S. conducted the experiments. D.K.B. performed the crystallographic analyses. J.S. oversaw the theoretical work, which R.S. carried out. Z.R.T. and D.D.S. performed and analysed the femtosecond transient absorption spectroscopy. M.W.P., R.S. and S.J.R. wrote the paper.
