## [Peer Review File · Nature Communications]

Reviewers' comments:

Reviewer #1 (Remarks to the Author):

This work shows very convincingly a relatively easy and plausibly prebiotic divergent access to the Watson-Crick-basepairing arabinonucleosides. From there, through prebiotic phosphorylation and dehydration/condensation reactions, a relatively straight-forward formation of mixed-sequence, stretch wise base-paired arabinonucleic acids (ANA) is imaginable.

If not for the fact that probably only RNA (theoretically also lyxose-based NA) have the prerequisite cis-diol configuration of 2',3'-hydroxy groups that are efficient in transferring aminoacyl esters to generate peptide bonds, ANA could indeed be the first genetic information carrier, being compatible with read-out by RNA or DNA. The question is whether translation preceded or followed replication, but this is out of the scope of the presented work.

From an experimental point of view this work is quite perfect, including the data presented in the supporting information.

Minor suggestions:

Table 1, caption: the reader should immediately realise that all yields have been determined by ¹H NMR spectroscopy at 600 MHz on the crude reaction mixtures.

Rather than "prebiotically plausible", "plausibly prebiotic".

Reviewer #2 (Remarks to the Author):

In this manuscript, Powner and co-workers report thorough investigations of different chemical routes for the conversion of anhydronucleosides to arabinonucleosides. After a comprehensive chemical survey of reactions focused on individual anhydronucleosides, the manuscript concludes with a one-pot reaction containing all the three relevant precursor anhydronucleosides, which goes on to produce all four arabinonucleosides functionalized with the canonical RNA nucleobases. The formation of side products at any stage appears to be minor. The authors note that the precursor of at least one other nucleoside (the arabinoinosine nucleoside) does not convert very efficiently, possibly suggesting a mechanism of selection for A, U, G, and C over other prebiotically available alternatives. The quantum chemistry and FTAS appear to corroborate this as well, and gives a satisfying explanation of why the sulfur atom can be detached photochemically.

The results presented in this manuscript are very interesting. The text is well written and easy to follow. A particular strength is the combination of experimental organic chemistry and theoretical photochemistry. The authors are bringing sulfur chemistry into their prebiotic synthesis of nucleotides, which is a welcome addition, as sulfur chemistry is probably under utilized in model prebiotic reactions compared to its likely involvement in the actual origins of life.

The results and conclusions of this manuscript will be of interest to a wide range of chemists, but to prebiotic chemists and nucleic acids chemists in particular. I am very supportive of publication, after some minor revisions:

Revisions that should be made before publication:

Major points to address:

1. The use of UV photons as part of the prebiotic nucleotide synthesis is certainly appropriate. Nevertheless, at least some semi-quantitative comparison should be presented in the main text for how the photon flux of the Rayonet RPR-200 reactor relates to the expected photon flux on the

prebiotic Earth. While it is not necessary that the photon flux be the same in laboratory experiments as is predicted to have been the proton flux on the prebiotic Earth, it would be reassuring if the flux required to drive the reaction over the course of the 16 hr experiments is within two or three orders of magnitude of what would have been incident on the Earth's surface. If the flux required is several orders of magnitude greater, then additional arguments the relevance of this photochemistry would need to be made.

2. Most of the NMR spectra presented were collected with a 9:1 H₂O:D₂O solvent (which is typical to provide a deuterium lock signal for the NMR, while also allowing detection of exchangeable protons). However, for some spectra, no resonances for protons attached to heteroatoms (i.e., chemically protic hydrogens) are shown. Are these resonances simply so far downfield that they are cutoff from the region shown? Do these protons exchange so fast their resonances are encompassed with the water peak? Are some of the NMR spectra actually collected with 100% D₂O? For example, there are no protic signals shown or stated in any spectra of compound 3C (either in the main text or SI), even though the text states that the spectra were taken in 9:1 H₂O:D₂O. Please check and add appropriate information to the text and/or figure captions where necessary.

3. Some justification, perhaps even a whole new paragraph, should be added about the apparent disparities in the oxidative/reductive nature of the environment that is required for this chemistry to proceed. On one hand, H₂S (i.e., sulfur in its most reduced state) is required to thiolize the anhydronucleosides, and on the other hand, H₂O₂ (an oxidizing agent) is required to push the thionucleoside species forward to their canonical forms. It seems unlikely that these two gaseous/volatile species could react sequentially and exclusively with the nucleoside substrates in this chemical scenario and not with each other to simply produce sulfur oxides/oxoacids. The author should also address, or at least speculate/propose, what environment could generate these two species that have such different redox potentials.

4. In the Introduction references 4, 5 and 6 are presented as literature examples of previous proposals of "plausible" (non-canonical) prebiotic nucleosides. These three references do not provide background adequate for the reader. Reference 4 is fully appropriate, as Eschenmoser has investigated possible alternatives to RNA in the prebiotic context. However, reference 5 by Taylor et al. demonstrate that alternative nucleic acids can evolve, but no information on the prebiotic relevance of these polymers is provided. Reference 6, by Sutherland and co-workers, reads more of a case against the idea that other nucleic acids came before RNA. Given the contributions to the idea that RNA was preceded by another polymer the Krishnamurthy and Hud laboratory, and the need to include a citation that gives adequate background on this topic, a good reference to add would be The Origin of RNA and 'My Grandfather's Axe', Hud et al., Chem. Biol. 20, 466-474, 2013.

5. The recent publication by the Krishnamurthy and Hud groups should also be mentioned and cited in the text, Glycosylation of a model proto-RNA nucleobase with non-ribose sugars: implications for the prebiotic synthesis of nucleosides, Fialho, et al., Org. Biomol. Chem. 16, 1263-1271, 2018. In this paper the authors report prebiotic synthesis of arabinose nucleosides with a plausible prebiotic nucleobase. The fact that arabinose nucleosides/nucleotides have now been prepared by two different synthetic routes strengthens the case for these nucleosides/nucleotides being present on the prebiotic Earth.

Minor edits

6. Page 3, paragraph 1: TNA is almost always used to abbreviate threose nucleic acid, not tetrose nucleic acid.

7. Page 22: The first paragraph of the conclusion contains the phrase "H₂O₂-oxidative is especially high yielding", which should probably read "H₂O₂-oxidation is especially high yielding".

8. Supplementary Figure 12: I believe the structure for ara-4I is incorrect. It has a methyl group, but shouldn't it be a hydroxymethyl group?

9. Supplementary Figures 75-85: Some structures are shown in their thiol tautomers, and others are shown in their thione tautomers, but in the main text, they are always shown in their thione tautomers. I suspect that the thione tautomers would dominate. Unless there is a good reason to depict them as their thiol tautomers, it may be better to change the structures to show them in their thione tautomers.

10. Supplementary Figures 121-122: The color scheme chosen for the atoms is unusual. Also, the captions for these figures list the color code incorrectly. Perhaps colors for nitrogen and oxygen have been accidentally reversed. Suggest changing figure to make the oxygen atoms red and the nitrogen atoms blue.

Reviewer #3 (Remarks to the Author):

Powner and colleagues studied the chemical conversion of anhydronucleosides to nucleosides. The authors treated the anhydronucleosides with hydrogensulfide, hydrogenperoxide or UV light in aqueous solution and determined the yield of the conversions. The reaction sequence usually consisted of two steps, the hydrolysis/thiolysis of the five-membered ring of the anhydronucleoside and functional group removal to obtain the free nucleosides. The nucleosides produced are arabino nucleosides, not natural ribo nucleosides. The starting materials were studied in related work by the authors and others. The authors find that the same reaction conditions produce pyrimidine and purine arabino nucleosides.

Major issues

- The title speaks of 'Selective Prebiotic Synthesis', and this reviewer expected to find full syntheses. Instead, the authors have started from anhydronucleosides, where the entire carbon framework is already assembled, and merely studied the steps mentioned above. This should be reflected in the title, which is too broad.
- As so often is the case in this field, it is not easy to see why these are 'prebiotic syntheses'. There is a series of steps involved, some of quite different type (strongly reducing conditions in the form of 0.67 M H₂S, then strongly oxidizing conditions in the form of H₂O₂, or forcing photochemical conditions in the form of 300 nm light; different pH values, heat or room temperature). It is not easy to see how molecules on primitive earth could have experience such a series of different conditions in the correct order to undergo the reactions observed.

Minor issues

- The paper is not well written und difficult to follow. For example, the section on the quantum chemical calculations is overly technical and barely connected to the remainder of the paper. The figure legends are a mix of legend and regular text plus conclusions and rather long. Table 1 is also difficult to follow and some entries appear in color. The term "pleasingly" appears too often, and some phrases are problematic, such as 'divergent prebiotic synthesis', 'important critical role', 'An efficient protocol ... would indicate', 'Watson-Crick base-pairing products'
- Three titles/subtitles in a row are unusual, e.g.

Results and Discussion

Purine Nucleoside Synthesis

8-Mercapto-purine synthesis [should also be mercaptopurine; no hyphen]

- The SI is overly long.

Response to Reviewers' Comments

We would like to thank all the reviewers for their time and constructive comments, and for the consequent improvements to our manuscript.

Reviewer #1:

Comment: This work shows very convincingly a relatively easy and plausibly prebiotic divergent access to the Watson-Crick-base pairing arabinonucleosides. From there, through prebiotic phosphorylation and dehydration/condensation reactions, a relatively straight-forward formation of mixed-sequence, stretch wise base-paired arabinonucleic acids (ANA) is imaginable. If not for the fact that probably only RNA (theoretically also lyxose-based NA) have the prerequisite cis-diol configuration of 2',3'-hydroxy groups that are efficient in transferring aminoacyl esters to generate peptide bonds, ANA could indeed be the first genetic information carrier, being compatible with read-out by RNA or DNA. The question is whether translation preceded or followed replication, but this is out of the scope of the presented work.

Response: Reviewer #1's comments are both insightful and interesting. Elucidating the chemical or biological process that first coupled information transfer between nucleic acids and peptides is undoubtedly a key element in understanding the origins of extant life. However, we agree with reviewer #1 that the origins of translation are beyond the scope of our current manuscript.

Comment: From an experimental point of view this work is quite perfect, including the data presented in the supporting information.

Response: We would like to thank reviewer #1 for their highly positive appraisal of our work and supporting information.

Comment: Table 1, caption: the reader should immediately realise that all yields have been determined by ¹H NMR spectroscopy at 600 MHz on the crude reaction mixtures.

Response: We have now noted in the table legend: "*Yields from the crude reaction were directly determined by ¹H NMR (600 MHz) spectroscopy.*"

Comment: Rather than "prebiotically plausible", "plausibly prebiotic".

Response: Although the term "prebiotically plausible" is widely used, we have nonetheless altered this term where it appeared in the manuscript.

"Prebiotically plausible" remains unchanged in the three reference titles that use this terminology.

Reviewer #2:

Comment: In this manuscript, Powner and co-workers report thorough investigations of different chemical routes for the conversion of anhydronucleosides to arabinonucleosides. After a comprehensive chemical survey of reactions focused on individual anhydronucleosides, the manuscript concludes with a one-pot reaction containing all the three relevant precursor anhydronucleosides, which goes on to produce all four arabinonucleosides functionalized with the canonical RNA nucleobases. The formation of side products at any stage appears to be minor. The authors note that the precursor of at least one other nucleoside (the arabinoinosine nucleoside) does not convert very efficiently, possibly suggesting a mechanism of selection for A, U, G, and C over other prebiotically available alternatives. The quantum chemistry and FTAS appear to corroborate this as well, and gives a satisfying explanation of why the sulfur atom can be detached photochemically.

The results presented in this manuscript are very interesting. The text is well written and easy to follow. A particular strength is the combination of experimental organic chemistry and theoretical photochemistry. The authors are bringing sulfur chemistry into their prebiotic synthesis of nucleotides, which is a welcome addition, as sulfur chemistry is probably under utilized in model prebiotic reactions compared to is likely involvement in the actual origins of life.

The results and conclusions of this manuscript will be of interest to a wide range of chemists, but to prebiotic chemists and nucleic acids chemists in particular. I am very supportive of publication, after some minor revisions:

Response: We thank reviewer #2 for their evaluation of our work.

Comment: The use of UV photons as part of the prebiotic nucleotide synthesis is certainly appropriate. Nevertheless, at least some semi-quantitative comparison should be presented in the main text for how the photon flux of the Rayonet RPR-200 reactor relates to the expected photon flux on the prebiotic Earth. While it is not necessary that the photon flux be the same in laboratory experiments as is predicted to have been the proton flux on the prebiotic Earth, it would be reassuring if the flux required to drive the reaction over the course of the 16 hr experiments is within two or three orders of magnitude of what would have been incident on the Earth's surface. If the flux required is several orders of magnitude greater, than additional arguments the relevance of this photochemistry would need to be made.

Response: These are insightful comments, providing the comparative flux ratio is indeed a useful metric for the reader and we are happy to provide this comparison. We can also reassure reviewer #2 that we are well within the parameters that they have outlined.

254 nm lamps: We used six of the available sixteen lamps in the Rayonet irradiation chamber. For six lamps power (P) = 3.9 (+/-0.3) mW = 3.9×10^4 erg/s; therefore, with a 1 cm² irradiation cross-section, flux = 3.9×10^4 erg/s/cm².

300 nm lamps: Again, we used six of the available sixteen lamps in the Rayonet irradiation chamber. P = 3.09 (+/-0.06) mW; therefore, with a 1 cm² irradiation cross-section, flux = 3.09×10^4 erg/s/cm².

Solar UV Flux: It is of note that UV light may have been the most-abundant source of energy available for prebiotic chemistry. The integrated 200-300 nm wavelength surface photon flux from early Sun is 2.7×10^3 erg/s/cm² (S. Ranjan and D. D. Sasselov, *Astrobiology* **16**, 68 (2016)). Therefore the irradiation flux in our experiments is approximately 14× or 11× that of the UV flux of early Sun at 254 nm and 300 nm, respectively. The experimental flux is only one order of magnitude above the predicted incident flux at the prebiotic Earth's surface.

We have added a comment to the discussion section, which compares the experimental and predicted Earth surface photon flux.

Comment: 2. Most of the NMR spectra presented were collected with a 9:1 H₂O:D₂O solvent (which is typical to provide a deuterium lock signal for the NMR, while also allowing detection of exchangeable protons). However, for some spectra, no resonances for protons attached to heteroatoms (i.e., chemically protic hydrogens) are shown. Are these resonances simply so far downfield that they are cutoff from the region shown? Do these protons exchange so fast their resonances are encompassed with the water peak? Are some of the NMR spectra actually collected with 100% D₂O? For example, there are no protic signals shown or stated in any spectra of compound 3C (either in the main text or SI), even though the text states that the spectra were taken in 9:1 H₂O:D₂O. Please check and add appropriate information to the text and/or figure captions where necessary.

Response: Although exchangeable proton resonances are commonly observed in protein and nucleic acid NMR spectroscopy, internal structure and hydrogen bonding can preclude rapid exchange in certain regions of these macromolecules, which enables some exchangeable proton resonances to be observed. The observation of signals from exchangeable proton resonances is a valuable method to ascertain which parts of these large structures are exposed to solvent (water). However, all proton resonances for the structures reported in this manuscript exchange rapidly on the NMR timescale and all exchangeable protons are exposed to solvent (water). Accordingly, no discrete exchangeable proton resonances are observed in 9:1 H₂O/D₂O. In aprotic hydrogen-bonding solvents, such as *d*₆-DMSO, exchangeable hydroxyl group resonances are observed (e.g. see Supplementary Fig. 83). However, it is not unusual that the exchangeable protons in small molecules (which are inevitably exposed to solvent water) are broadened by exchange and average with the HOD signal, and therefore we did not feel this warranted comment. Nevertheless, we have now added a comment to the General Experimental section in the supporting information that notes:

Rapidly exchanging proton (O-H, N-H) resonances are not detected due to signal broadening and coalescence with the HOD signal in 9:1 H₂O/D₂O.

Comment: 3. Some justification, perhaps even a whole new paragraph, should be added about the apparent disparities in the oxidative/reductive nature of the environment that is required for this chemistry to proceed. On one hand, H₂S (i.e., sulfur in its most reduced state) is required to thiolize the anhydronucleosides, and on the other hand, H₂O₂ (an oxidizing agent) is required to push the thionucleoside species forward to their canonical forms. It seems unlikely that these two gaseous/volatile species could react sequentially and exclusively with the nucleoside substrates in this chemical scenario and not with each other to simply produce sulfur oxides/oxoacids. The author should also address, or at least speculate/propose, what environment could generate these two species that have such different redox potentials.

Response: We thank reviewer #2 for this suggestion. Whilst the focus of this manuscript has been to develop new chemistry, not make an assessment of geochemical boundary conditions, we have carefully considered whether the reported chemistry could reasonably take place under conditions that would not violate environments the planetary research community accepts as plausible on early Earth. Accordingly, we have willingly added a new paragraph to the discussion section of the paper to clarify our thoughts with respect to H₂S and H₂O₂.

Comment: 4. In the Introduction references 4, 5 and 6 are presented as literature examples of previous proposals of "plausible" (non-canonical) prebiotic nucleosides. These three references do not provide background adequate for the reader. Reference 4 is fully appropriate, as Eschenmoser has investigated possible alternatives to RNA in the prebiotic context. However, reference 5 by Taylor et al. demonstrate that alternative nucleic acids can evolve, but no information on the

prebiotic relevance of these polymers is provided. Reference 6, by Sutherland and co-workers, reads more of a case against the idea that other nucleic acids came before RNA. Given the contributions to the idea that RNA was preceded by another polymer the Krishnamurthy and Hud laboratory, and the need to include a citation that gives adequate background on this topic, a good reference to add would be The Origin of RNA and 'My Grandfather's Axe', Hud et al., Chem. Biol. 20, 466-474, 2013.

Response: a) Firstly, we note that we did not specify we were discussing “plausible” **non-canonical** nucleotide candidates; reviewer #2 has inserted the term non-canonical and changed the meaning of our sentence. To be clear, we are discussing “plausible” nucleotide candidates, and we openly specify that these include RNA (see manuscript extract highlighted in bold below).

*“... many “plausible” nucleoside candidates have been suggested to have played a role at the origins of life,⁴⁻⁶ the concurrent prebiotic synthesis of a complete set of nucleoside monomers remains an unresolved challenge for any of the proposed genetic polymers [e.g. **ribonucleic acid (RNA)**, arabinonucleic acid (ANA), threonnucleic acid (TNA), pyranosyl-ribonucleic acid (pRNA)]”*

We understand very well the content and context of reference 6. The corresponding author also co-authored this review. As noted above, we are discussing all plausible nucleic acids including RNA. However, reference 6 does more than acknowledge the RNA-first hypothesis, it contextualises the difficulties faced in selective prebiotic XNA syntheses and their onward evolution to extant RNA-based biology (using RNA, pRNA, GNA and TNA as case studies). This makes reference 6 a very well justified reference in the context it is used.

Moreover, although reference 6 makes a strong case against XNAs that vary significantly from RNA structure, ANA and RNA only differ by a single stereochemical change at the C2' carbon atom of their respective monomers.

Importantly, in this current manuscript, we demonstrate that the conceptual difficulties outlined in reference 6 can be overcome for prebiotic ANA nucleoside synthesis. For example, it is of note that reference 6 provides a detailed discussion of stereoselective prebiotic sugar syntheses in the context of XNA's, and presents a stereochemical model to explain why prebiotic oxazoline synthesis favours *ribo*- and *arabino*-furanosyl products. The stereochemical argument made in reference 6, in support of RNA, could now also be used to support ANA.

b) We thank reviewer #2 for noting that we could provide more background references to contextualise the debate of nucleic acids from the perspective of prebiotic chemistry. We have now included both recommended Krishnamurthy/Hud references, as well as:

7. Fuller, W. D., Sanchez, R. A. & Orgel, L. E. Studies in Prebiotic Synthesis. VI. Synthesis of Purine Nucleosides. J. Mol. Biol. 67, 25–33 (1972).

8. Joyce, G. F., Schwartz, A. W., Miller, S. L. & Orgel, L. E. The Case for an Ancestral Genetic System Involving Simple Analogues of the Nucleotides. Proc. Natl. Acad. Sci. U. S. A. 84, 4398–402 (1987).

9. Eschenmoser, A. & Loewenthal, E. Chemistry of Potentially Prebiological Natural Products. Chem. Soc. Rev. 21, 1–16 (1992).

10. Böhler, C., Nielsen, P. E. & Orgel, L. E. Template Switching Between PNA and RNA Oligonucleotides. Nature 376, 578–581 (1995).

11. Eschenmoser, A. Chemical Etiology of Nucleic Acid Structure. Science 284, 2118–2124 (1999).

12. Nielsen, P. E. Peptide Nucleic Acid. A Molecule with Two Identities. Acc. Chem. Res. 32, 624–630 (1999).

13. Schöning, K.-U. et al. Chemical Etiology of Nucleic Acid Structure: the α -Threofuranosyl-(3'→ 2') Oligonucleotide System. *Science* 290, 1347–1351 (2000).
14. Orgel, L. E. Prebiotic Adenine Revisited: Eutectics and Photochemistry. *Orig. Life Evol. Biosph.* 34, 361–369 (2004).
15. Zhang, L., Peritz, A. & Meggers, E. A Simple Glycol Nucleic Acid. *J. Am. Chem. Soc.* 127, 4174–4175 (2005).
17. Sutherland, J. D. Ribonucleotides. *Cold Spring Harb. Perspect. Biol.* 2, a005439 (2010).
18. Powner, M. W. & Sutherland, J. D. Prebiotic Chemistry: A New Modus Operandi. *Philos. Trans. R. Soc. B Biol. Sci.* 366, 2870–2877 (2011).
19. Rios, A. C. & Tor, Y. On the Origin of the Canonical Nucleobases: An Assessment of Selection Pressures Across Chemical and Early Biological Evolution. *Isr. J. Chem.* 53, 469–483 (2013).
22. Kim, H.-J. & Benner, S. A. Prebiotic Stereoselective Synthesis of Purine and Noncanonical Pyrimidine Nucleotide from Nucleobases and Phosphorylated Carbohydrates. *Proc. Natl. Acad. Sci.* 114, 11315–11320 (2017).

Comment: 5. The recent publication by the Krishnamurthy and Hud groups should also be mentioned and cited in the text, Glycosylation of a model proto-RNA nucleobase with non-ribose sugars: implications for the prebiotic synthesis of nucleosides, Fialho, et al., *Org. Biomol. Chem.* 16, 1263-1271, 2018. In this paper the authors report prebiotic synthesis of arabinose nucleosides with a plausible prebiotic nucleobase. The fact that arabinose nucleosides/nucleotides have now been prepared by two different synthetic routes strengthens the case for these nucleosides/nucleotides being present on the prebiotic Earth.

Response: We have cited this paper, as requested.

We also strongly agree that; “*two different synthetic routes [to ANA] strengthens the case for these nucleosides/nucleotides being present on the prebiotic Earth*”. Indeed, we have previously made the same point for prebiotic RNA synthesis (Islam & Powner *Chem* 2, 470 (2017)), because the canonical pyrimidine ribonucleotides can both be synthesised by two different prebiotic mechanisms from two diastereomeric starting materials (*i.e. ribo-* and *arabino-*aminooxazoline; see Powner et al *Nature* 459, 239–242 (2009) and Xu et al. *Nat. Chem.* 9, 303 (2017), respectively).

Therefore, it is of note that in this manuscript we demonstrate two mechanisms for mercaptopurine reduction that operate under different prebiotically plausible conditions. Accordingly, we have now noted in our discussion, that:

“... *two distinct mercaptopurine (5) reductions demonstrate not only the specific value of sulfide in prebiotic nucleoside synthesis but also, more generally, the value of chemical redundancy. Two mechanisms for mercaptopurine reduction, that operate under different conditions but both furnish the same purine products, renders the overall transformation more robust. This should be a general chemical principle and may be especially important to consider in the context of the origins of life. Chemical redundancy is highly likely to improve pathway or network robustness, which may be an essential feature of sustained proto-metabolism in a (geo)chemically fluctuating environment.*”

However, although we agree that (chemical) network redundancy provides robustness, and “*strengthens the case for these nucleosides/nucleotides being present on the prebiotic Earth*”, it would be disingenuous to imply the same ANA nucleosides are made in the Krishnamurthy/Hud

synthesis. Although interesting, the mechanism reported by Krishnamurthy/Hud is not specific to either arabinosylation or furanosyl-nucleoside synthesis, and, importantly, in the context of our current manuscript, it is not relevant to the biological Watson-Crick nucleobases.

Comment: 6. Page 3, paragraph 1: TNA is almost always used to abbreviate threose nucleic acid, not tetrose nucleic acid.

Response: Corrected to *threonucleic acid*.

Comment: 7. Page 22: The first paragraph of the conclusion contains the phrase "H₂O₂-oxidative is especially high yielding", which should probably read "H₂O₂-oxidation is especially high yielding".

Response: Corrected.

Comment: 8. Supplementary Figure 12: I believe the structure for ara-4I is incorrect. It has a methyl group, but shouldn't it be a hydroxymethyl group?

Response: Corrected.

Comment: 9. Supplementary Figures 75-85: Some structures are shown in their thiol tautomers, and others are shown in their thione tautomers, but in the main text, they are always shown in their thione tautomers. I suspect that the thione tautomers would dominate. Unless there is a good reason to depict them as their thiol tautomers, it may be better to change the structures to show them in their thione tautomers.

Response: This is a reasonable request. We too suspect the thiones will dominate at tautomeric equilibrium. Moreover, X-ray crystallography (in the solid state) and solution NMR spectroscopy both indicate these structures predominantly exist as their thione tautomers. Therefore, as requested, those structures that appeared as the thiol tautomer in the supplementary information document have been redrawn in their thione tautomeric form.

Comment: 10. Supplementary Figures 121-122: The color scheme chosen for the atoms is unusual. Also, the captions for these figures list the color code incorrectly. Perhaps colors for nitrogen and oxygen have been accidentally reversed. Suggest changing figure to make the oxygen atoms red and the nitrogen atoms blue.

Response: The colour scheme used in figures 121 – 122 is correct; the figure images remain unaltered (oxygen = red, nitrogen = blue). However, we have modified the figure legends to clarify the colour scheme used. We acknowledge that the use of hyphens and commas may have been confusing and we have replaced the hyphens with equals signs to remove any ambiguity:

Colour scheme: carbon = grey, hydrogen = white, nitrogen = blue and oxygen = red.

Reviewer #3 (Remarks to the Author):

Comment: Powner and colleagues studied the chemical conversion of anhydronucleosides to nucleosides. The authors treated the anhydronucleosides with hydrogensulfide, hydrogenperoxide or UV light in aqueous solution and determined the yield of the conversions. The reaction sequence usually consisted of two steps, the hydrolysis/thiolysis of the five-membered ring of the

anhydronucleoside and functional group removal to obtain the free nucleosides. The nucleosides produced are arabino nucleosides, not natural ribo nucleosides. The starting materials were studied in related work by the authors and others. The authors find that the same reaction conditions produce pyrimidine and purine arabino nucleosides.

Response: No specific response is required to this summary of some of our reported work. However, we would note that we have also demonstrated reactions that produce “*natural*” pyrimidine and purine “*ribo nucleosides*”.

We report the synthesis of canonical pyrimidines, β -uridine (β -*ribo-4U*; 78%) and β -cytidine (β -*ribo-4C*; quant.). These are thought to be prebiotic, building on the work of Sutherland and co-workers (reference 33; Xu et al *Nat. Chem.* **9**, 303–309 (2017)).

We also report the selective photochemical reduction of β -*ribo*-mercaptapurines to afford their respective canonical purine nucleotides: β -adenosine (β -*ribo-4A*; 70%), β -guanosine (β -*ribo-4G*; 80%) and β -inosine (β -*ribo-4I*; 10%). However, a prebiotic synthesis of β -*ribo*-mercaptapurines has not yet been reported.

Comment: The title speaks of 'Selective Prebiotic Synthesis', and this reviewer expected to find full syntheses. Instead, the authors have started from anhydronucleosides, where the entire carbon framework is already assembled, and merely studied the steps mentioned above. This should be reflected in the title, which is too broad.

Response: We strongly disagree. The title used is an accurate description of the key theme of the manuscript. It conveys clearly the manuscript's conclusions and its key experimental and conceptual advances. This paper develops a novel prebiotic synthesis of ANA nucleosides – as clearly expressed by the title.

With respect to the “assembled carbon framework”, one could argue similarly for any manuscript that used, for example, a preformed sugar or nucleobase for nucleoside synthesis; clearly this is an untenable argument if there are reported prebiotic syntheses of the precursors. Scientific advance is made incrementally, and one is not expected to disconnect to one-carbon atom units to complete every “synthesis”. One is certainly not expected to re-publish (prebiotic) syntheses. A synthesis begins from available or reasonable (prebiotic) precursors, and this is precisely what we have done. It is of note that by building on our previous work (e.g. Powner et al *Nature* 2009, Stairs et al *Nat. Commun.* 2017) a complete, contiguous and selective prebiotic synthesis of both purine and pyrimidine ANA nucleosides has now been elucidated.

Comment: As so often is the case in this field, it is not easy to see why these are 'prebiotic syntheses'. There is a series of steps involved, some of quite different type (strongly reducing conditions in the form of 0.67 M H₂S, then strongly oxidizing conditions in the form of H₂O₂, or forcing photochemical conditions in the form of 300 nm light; different pH values, heat or room temperature). It is not easy to see how molecules on primitive earth could have experience such a series of different conditions in the correct order to undergo the reactions observed.

Responses: We fervently disagree. It is easy to envisage molecules experiencing different conditions on the early Earth.

H₂S and H₂O₂: It would be a most unusual, and almost certainly inaccurate, view of the early Earth to believe that there was only one redox state available in all early Earth environments. The whole planet is at redox disequilibrium, and likely has been since the differentiation of the Earth's core. All reagents used are plausibly prebiotic, and although different redox conditions are implicated, this is not geochemically implausible. Layer redox gradients are geochemically common and it is not difficult to imagine compounds crossing a redox gradient. The mantle's oxygen fugacity likely remains similar to the prebiotic era (Trail et al *Nature* **480**, 79–82 (2011)),

and sulfide leaching and volcanic H₂S outgassing must have occurred on the early Earth (Ranjan et al. *arXiv Prepr.* 1–44 (2018)). Photo-dissociation generates hydrogen peroxide from water, and H₂O₂ was likely to be a key environmental oxidant prior to global oxidation (Kasting, J. F., Holland, H. D., Pinto, J. P. *J Geophys. Res. Atmos.* **90**, 497–510 (1985) and Catling, D. C. & Kasting, J. F. *Atmospheric Evolution on Inhabited and Lifeless Worlds*. (Cambridge University Press, 2017)). Therefore, it is certainly conceivable that a redox gradient could be generated across plausible geochemical environments. However, although this geochemistry is unquestionably interesting, it is also clearly outside the scope of our current manuscript.

With respect to ordering: Although the reaction with H₂S and then H₂O₂ must occur in sequence to achieve purine reduction by the second mechanism described in our manuscript, it is of note that the anhydronucleosides are stable to H₂O₂ oxidation (and incidentally photo-reduction) and the reduced purines are stable to H₂S. Therefore cycling material between these redox conditions or across this redox gradient would not be problematic for this chemistry.

It is also of note that purine reduction by thiolysis and photochemical irradiation does not require two redox states, only hydrogen sulfide and UV light, both of which are expected to be in adequate (simultaneous) supply on the early Earth (Ranjan et al. *arXiv Prepr.* 1–44 (2018)).

Heat: The temperatures exploited in our reactions (20 – 60 °C) are well within bounds considered to be reasonable temperatures on the early Earth. It is very easy to envisage fluctuating temperature on the early Earth, include those due to day-night cycles. It is also appealing to consider hot springs, fumaroles and mineral (sulfur) springs as the source of both temperature and H₂S gradients. However, although these considerations are certainly interesting, investigating geology is, again, beyond the scope of our current manuscript.

“Different” pH values: Reviewer #3 suggests that “different pH values” are required, but this is not accurate.

We have explored different pH values to be chemically thorough, not because they are prebiotically necessary. All prebiotic reactions are investigated and reported between pH 6.5 – 7.5. Any reaction that deviates from neutral pH (pH < 6 or pH > 8) is not a required prebiotic reaction step, and forms part of the broader chemical context of the investigation. All high/low pH conditions are either for safety (when removing H₂S from reactions) or to demonstrate an interesting point of reactivity, which may be beyond the scope of prebiotic chemistry, but of interest to the chemical community.

Hydrogen sulfide is degassed (a safety measure, when handling this extremely toxic gases) following reaction completion and prior to manipulation outside a ventilated fume cupboard or sealed vessel. This degassing is most efficient at a pH significantly below the compound's pK_a because H₂S (not NaHS) is volatile. Accordingly, to maximise the safety of these solutions/reactions, H₂S is degassed at low pH, but lowering the pH is not prebiotically essential and we have not suggested this is prebiotic. However, although not prebiotically essential, we would recommend in the strongest possible terms that anyone wishing to repeat our work or use H₂S for other purposes take extreme care when handling H₂S (and its conjugate base) because it is extremely toxic and can be lethal (LC₅₀=800 ppm).

The rearrangement of purine anhydronucleosides was carried out at high pH, but is not described as a “prebiotic reaction”; however, this reaction may be of interest to the broader chemical community. It was carried out to demonstrate the remarkable stability of these anhydronucleosides to hydrolysis. We don't suggest this high pH reaction is necessary or prebiotically relevant, but it was interesting to observe **3G** isomerise to its 3,3'-isomer at high pH.

C2-Mercapto-pyrimidine reduction (which would incidentally be detrimental to the synthesis of canonical pyrimidine nucleobases) is only observed at low pH. We don't suggest this reaction is prebiotic either, in fact we specifically note that under prebiotically plausible conditions (*i.e.* buffered at neutral pH) this reaction does not occur. Moreover, we note that C2-mercapto-

pyrimidine is not a significant product of aqueous hydrolysis (its synthesis requires dried solvents). We note that the low pH of this pyrimidine reduction is an important difference between purine and pyrimidine reactivity. Indeed this difference may be essential to furnishing Watson-Crick base pairing purines and pyrimidine nucleosides simultaneously under prebiotically plausible conditions. This low pH reaction exemplifies the switch in purine and pyrimidine reactivity, but is not an essential reaction step in ANA nucleoside synthesis.

“Forcing” photochemical conditions: UV light may have been the most-abundant source of energy available for prebiotic chemistry, and our photon flux is only about 11× the predicted UV photon flux on the early Earth’s surface. We have limited the photon flux in our experiments to a reasonable level. We have now added a statement to the manuscript to specify the relationship between the applied Rayonet flux and the predicted solar flux on the early Earth. Moreover, whilst we have used monochromatic emission (either 254 nm or 300 nm) in these reactions, it is expected that photochemical conditions on the early Earth would have been capable of supplying the necessary photon flux. The early Sun outputs significant 200-300 nm UV light, especially at the longer wavelength (>250 nm) in this range (see S. Ranjan and D. D. Sasselov, *Astrobiology* **16**, 68 (2016)), which is inline with our reaction conditions. As we have observed that photoreduction is efficient at both 254 and 300 nm, we fully expect that 200-300 nm irradiation (and especially the more intense >250 nm radiation from the early Sun) would be capable of driving these reactions. Of course, there might be some wavelength dependence to the efficiency of the photoreduction, and this is of potential interest, but is out of the scope of our current manuscript.

General prebiotic plausibility: Prebiotic plausibility has been addressed, for example, by Leslie Orgel and Albert Eschenmoser, who have both made seminal contributions to this field: **a)** Orgel, L. E. *Crit. Rev. Biochem. Mol. Biol.* **39**, 99–123 (2004). **b)** Eschenmoser, A. *Tetrahedron* **63**, 12821–12844 (2007). These reviews are both cited in our manuscript.

a) Orgel presents three principles that must be met for a reaction to be considered prebiotic: **1.** *It must be plausible, at least to the proposers of a prebiotic synthesis, that the starting materials for a synthesis could have been present in adequate amounts at the site of synthesis.* **2.** *Reactions must occur in water or in the absence of a solvent.* **3.** *The yield of the product must be significant.*

Our conditions meet all the criteria specified by Orgel.

b) Eschenmoser recognises: *Uncertainty about the physical, chemical, and locational boundary conditions of life's origin will persist. This is why biogenesis, as a problem of science, is lastly going to be a problem of synthesis. The origin of life cannot be ‘discovered’, it has to be ‘re-invented’.*

We ardently agree with Eschenmoser and our methodology aligns with the essential principle of *synthetic discovery* Eschenmoser declared must be the key to solving the problem of the origins of life. Moreover, these principles and our reactions are in accordance with our published views on using systems chemical analysis and chemical synthesis to drive prebiotic discovery (Powner & Sutherland *Phil. Trans. R. Soc. B* **366**, 2870–2877 (2011)).

Comment: The paper is not well written und difficult to follow, the section on the quantum chemical calculations is overly technical and barely connected to the remainder of the paper.

Response: We strongly disagree. The quantum chemical calculations are an integral part of this manuscript and directly link to the photochemical reduction of mercaptopurines, which is discussed in the preceding section of the paper. They are also implicitly linked to the ultrafast (femtosecond) transient absorption spectroscopy (FTAS) in the following section of the paper. The quantum chemical calculations are further discussed with respect to the mechanism of H₂O₂ oxidation, and the C8-purine/C2-pyrimidine carbene intermediates, which are predicted to facilitate nucleobase reduction under both photochemical and oxidative conditions.

We recognise that this manuscript is multidisciplinary, and that the technically accurate description of the work will vary across these disciplines. However, we agree with reviewer #2; “*a particular strength is the combination of experimental organic chemistry and theoretical photochemistry*”. Reviewer #2 also noted “*the text is well written and easy to follow.*”

We do not believe the computational section was overly technical; the details given are required to explain to the reader the results of the computation study. The numerical values are essential for comparison and the description of singlet and triplet states are inherent to the nature of the chemistry under investigation. Nevertheless we have made changes to this section of the manuscript to help guide readers through these results.

Comment: The figure legends are a mix of legend and regular text plus conclusions and rather long.

Response: We have prepared our legends following the criteria specified by *Nature*. The figure legends are all within the guideline length (350 words). They begin with a brief title for the whole figure, and continue with a statement of what is depicted in the figure, describing each panel. Legends are detailed enough so that each figure and caption can be understood in isolation from the main text, as requested by *Nature*.

Comment: Table 1 is also difficult to follow and some entries appear in color.

Response: The table closely reflected similar tables that have been published in *Nature* journals (e.g. Patel et al *Nat. Chem.* **7**, 301–307 (2015)). The table reports yield vs. conversion, drawing together the various results in the manuscript into one table to facilitate comparison.

We have aligned the conversions to ensure each row reports sequential conversions. We have also regrouped the conversions by nucleobase then stereochemistry, although this makes the table less space efficient, it could help the readers follow relationship between the yields reported in Table 1.

The table legend clearly states: “*arabino-Nucleoside yields highlighted in blue.*” The conversions have been highlighted in blue to help the reader follow which transformations yield ANA nucleosides.

Comment: The term “pleasingly” appears too often, and some phrases are problematic, such as ‘divergent prebiotic synthesis’, ‘important critical role’, ‘An efficient protocol ... would indicate’, ‘Watson-Crick base-pairing products’.

Response: It was not our intention use the word “pleasingly” too often. Although this was a fair description of our feeling on observing the reported results, we have reduced the number of times we used the word “pleasingly”. We have also deleted the word “important”, this was superfluous, but we cannot see a problem with any of the other terms specified. These statements are accurate to the concept we wish to convey and reflect the results that we report.

Comment: Three titles/subtitles in a row are unusual, e.g. Results and Discussion Purine Nucleoside Synthesis 8-Mercapto-purine synthesis

Response: We have removed the subtitle “purine nucleoside synthesis”.

Comment: 8-Mercapto-purine synthesis should also be mercaptopurine; no hyphen

Response: Changed.

Comment: The SI is overly long.

Response: We disagree very strongly with the sentiment of this comment.

The length of the SI is proportional to the number of unique experiments undertaken. We provided the data required to verify all results and the spectra that would be required to verify repetition of our experiment in other laboratories.

The length, detail and style of the SI document is comparable our recently published *Nat. Commun.* supplementary information, which was commended as “*the new Gold Standard in Supplementary Information documentation*”. Please see peer review file (page 15):

<https://media.nature.com/original/nature-assets/ncomms/2017/170519/ncomms15270/extref/ncomms15270-s2.pdf>

Finally, we also note that reviewer #1 stated: “*From an experimental point of view this work is quite perfect, including the data presented in the supporting information*”. Accordingly, we have not removed the reactions, details, spectra or characterisation from our SI document.

Reviewers' comments:

Reviewer #1 (Remarks to the Author):

The author's additions in the revised manuscript, viz. those that have been provoked by the response of Reviewer #2, are pleasingly comforting and plausibly enhance, for the general reader, the clarity of some of their most important statements. It is of note that Reviewer #1 is fervently opposed to the main criticism of Reviewer #3.

Reviewer #2 (Remarks to the Author):

I have gone through the comments/critiques of the three referees and the responses/edits provided the authors. I am mostly satisfied with the revised manuscript. I did, however, find one inconsistency between the edits reportedly made by the authors and the revised manuscript, which might be due to uploading an incorrect version of the manuscript.

The latter part of Comment 4 of Reviewer 2 reads:

Given the contributions to the idea that RNA was preceded by another polymer the Krishnamurthy and Hud laboratory, and the need to include a citation that gives adequate background on this topic, a good reference to add would be The Origin of RNA and 'My Grandfather's Axe', Hud et al., Chem. Biol. 20, 466-474, 2013.

The authors response to this point is:

b) We thank reviewer #2 for noting that we could provide more background references to contextualise the debate of nucleic acids from the perspective of prebiotic chemistry. We have now included both recommended Krishnamurthy/Hud references, as well as:

Contrary to this statement, a citation to the paper The Origin of RNA and 'My Grandfather's Axe', Hud et al., Chem. Biol. 20, 466-474, 2013, is still lacking from the revised manuscript. This citation should be added to provide the readers with a more recent review of past proposals for nucleic acids that may have come before RNA, as the current manuscript is proposing that earlier nucleic acids may have had a sugar other than ribose.

Reviewer #3 (Remarks to the Author):

This reviewer suggested major revision. Minor revisions were made. What is reported is an incremental advance, not a full new synthesis, and some yields are quite low. The procedures require many manual steps, and reagents such as 30% H₂O₂ are unrealistic prebiotically. It is not easy to recommend this manuscript for publication in a Nature journal.

Some of the many issues that remain are listed below.

- The title continues to be misleading.

The nucleosides that were synthesized from their anhydro precursors are not 'Watson-Crick nucleosides'. They do not contain ribose. [Perhaps one should also mention that Watson and Crick proposed a model for DNA double helices, not the structure of nucleosides, so the term itself is problematic.]

To call a two-step conversion of a fully assembled anhydronucleoside to a nucleoside a 'synthesis' without specifying what the advanced starting materials is, is problematic.

The remainder of the manuscript also continues to contain problematic language.

- If this is a synthesis (as the title suggests), the paper should report proper yields (of isolated materials). Instead, the authors call the conversion to products, detected in a mixture by spectroscopic techniques, 'yields'. This is inappropriate for chemists.

- 'Table 1| Tabulated reaction yields. Yields from the crude reaction were directly determined by ¹H NMR (600 MHz) spectroscopy.'

There is no need to state that a table contains tabulated data. Furthermore, neither 'reaction yields' nor 'yields from crude reactions' are defined. Yields are determined after a procedure inducing a reaction has been completed. While the reaction occurs, there is no yield. The term 'crude product' is defined. 'Crude reaction' is not.

- 'Accordingly, to address this problem we set out to develop new chemistry.^{1–4,18}'

If this is 'new chemistry', why are there five references for it? If this synthesis re-enacts prebiotic evolution, why does 'new chemistry' (a poorly defined term) have to be developed? If this is prebiotic chemistry, it is not new.

Supplementary Information

- 'The reaction was then cooled to RT.' (General procedure A).

A reaction is a process and cannot be cooled. A solution can be cooled or an apparatus, but not a process.

Likewise, 'After 2 d, the solution had risen to pH 9.6' makes no sense. The pH of a solution can rise. A solution cannot rise (unless one believes in magic).

- The nomenclature in the SI is confusing. (e.g. Arabino-2',2-Anhydrocytidine 3C Hydrolysis in the presence of 2',8-Anhydro-Adenosine 3A). This is neither a sentence nor a title. Proper nomenclature would be 2',8-anhydroadenosine (one word, lower case, no hyphen). Also, note the typographical error (presence), easily detected by a spell checker.

- At least in standard scientific English, 'Spectrum to show the ... products' should be 'spectrum showing the products of .. '

- When the authors write 'RT', they probably mean room temperature (r.t.) not gas constant times absolute temperature (RT).

- What is a 'lyophilite'?

- What is 'Pyrimidine UV Irradiation'? (Irradiation of pyrimidines?) What are 'Pyrimidine Oxidations'?

- Sometimes it is not clear what was done. For example, '...white solid. These solids were dissolved in boiling H₂O (40.0 mL) and allowed to cool to RT. After 24 h precipitation of crystalline solid was observed. These solids were isolated by filtration to afford 31.2 mg of 2',8-anhydroinosine 3I (31%) as off-white translucent crystals.' (Solids that are dissolved cannot be allowed to cool because they are no longer solids. Also, the switching between singular and plural makes this section more than confusing. It sounds as if several fractions [batches] were combined)

- The general part and some figure legends give the pulse program for the acquisition of one-dimensional proton NMR spectra as "noesygppr1d". Why is this pulse program being used? The title suggests that it is for NOESY spectra, not for 1D spectra (typically called "zg" in Bruker

parlance).

- Why are NMR data listed for known compounds? Repeatedly listing such data, following a statement "Experimental data is consistent with the literature" just bogs down the reader and is inconsistent with good scientific practice.

The authors proudly declare in their rebuttal letter that their SI format is now the "gold standard". In this reviewer's humble opinion, repeating data from the literature is a waste of space, even in the SI.

Response to Reviewers' Comments

Reviewer #1:

Comment: The author's additions in the revised manuscript, viz. those that have been provoked by the response of Reviewer #2, are pleasingly comforting and plausibly enhance, for the general reader, the clarity of some of their most important statements. It is of note that Reviewer #1 is fervently opposed to the main criticism of Reviewer #3.

Response: We thank reviewer #1 for their comments.

Reviewer #2:

Comment: I have gone through the comments/critiques of the three referees and the responses/edits provided the authors. I am mostly satisfied with the revised manuscript. I did, however, find one inconsistency between the edits reportedly made by the authors and the revised manuscript, which might be due to uploading an incorrect version of the manuscript ... a citation to the paper The Origin of RNA and 'My Grandfather's Axe', Hud et al., Chem. Biol. 20, 466-474, 2013, is still lacking from the revised manuscript.

Response: We are grateful to reviewer #2 for bringing this to our attention. This has been corrected and it is now reference 20.

Reviewer #3:

Comment: This reviewer suggested major revision. Minor revisions were made. What is reported is an incremental advance, not a full new synthesis, and some yields are quite low. The procedures require many manual steps, and reagents such as 30% H₂O₂ are unrealistic prebiotically.

Response: **a)** We are grateful for all the reviewers suggested improvements to our manuscript and we have complied with all reviewer suggestions or provided a detailed response and justification to their questions and comments. **b)** The optimised *arabino*-nucleoside conversions are >66% and many conversions are near-quantitative, these are good to excellent yields. **c)** Dilute H₂O₂ is prebiotically plausible. Dilute H₂O₂ was used in the reported reactions (e.g. General procedure C is reported with 150 mM H₂O₂ not 9.8 M H₂O₂). We have provided detailed discussion outlining the prebiotic plausibility of H₂O₂. **d)** Although certainly interesting, alternatives mixing methods (e.g. flow mixing) are beyond the scope of our current manuscript.

Comment: The title continues to be misleading. The nucleosides that were synthesized from their anhydro precursors are not 'Watson-Crick nucleosides'. They do not contain ribose. [Perhaps one should also mention that Watson and Crick proposed a model for DNA double helices, not the structure of nucleosides, so the term itself is problematic.] To call a two-step conversion of a fully assembled anhydronucleoside to a nucleoside a 'synthesis' without specifying what the advanced starting materials is, is problematic.

Response: We have made all recommended changes to the title. We have removed "synthesis" and specifying the starting materials are "anhydronucleosides" as requested. We have also changed "Watson-Crick" to "Watson-Crick base-pairing" which is widely preceded for nucleoside structures other than DNA [e.g. Eschenmoser, A. Chemical Etiology of Nucleic Acid Structure. *Science* **284**, 2118-2124 (1999)].

Comment: If this is a synthesis (as the title suggests), the paper should report proper yields (of isolated materials). Instead, the authors call the conversion to products, detected in a mixture by spectroscopic techniques, 'yields'. This is inappropriate for chemists.

Response: **a)** We have changed the title as requested (see above). **b)** We have also revised the text as requested, changing “yield” to “conversion” for spectroscopically determined yields. Whereas when isolated yields are reported (e.g. *ara-4I*) we have used the term “yield” as recommended by reviewer #3.

Comment: - 'Table 1| Tabulated reaction yields. Yields from the crude reaction were directly determined by 1H NMR (600 MHz) spectroscopy.' There is no need to state that a table contains tabulated data. Furthermore, neither 'reaction yields' nor 'yields from crude reactions' are defined. Yields are determined after a procedure inducing a reaction has been completed. While the reaction occurs, there is no yield. The term 'crude product' is defined. 'Crude reaction' is not.

Response: We have deleted “tabulated” and clarified nucleoside “conversions” are reported in Table 1. We have also specified the conversions in Table 1 were directly determined in the “crude product mixture” (rather than crude reaction) as recommended by reviewer #3.

Comment: 'Accordingly, to address this problem we set out to develop new chemistry.1–4,18' If this is 'new chemistry', why are there five references for it? If this synthesis re-enacts prebiotic evolution, why does 'new chemistry' (a poorly defined term) have to be developed? If this is prebiotic chemistry, it is not new.

Response: We thank reviewer #3 for bringing this to our attention. These cross-references were made to review articles outlining the need to develop “new chemistry” to elucidate the origins of life, but we thank reviewer #3 for noting the potential confusion this might cause. We have accordingly deleted these. We have also changed “chemistry” to “chemical reactions”, rephrased “develop” to “elucidate” and removed the word “new” as recommended.

Supplementary Information

Comment: 'The reaction was then cooled to RT.' (General procedure A).

A reaction is a process and cannot be cooled. A solution can be cooled or an apparatus, but not a process.

Response: We have changed “reaction” to “solution”.

Comment: Likewise, 'After 2 d, the solution had risen to pH 9.6' makes no sense. The pH of a solution can rise. A solution cannot rise (unless one believes in magic).

Response: We have changed “solution” to “pH of the solution”.

Comment: The nomenclature in the SI is confusing. (e.g. Arabino-2',2-Anhydrocytidine 3C Hydrolysis in the presence of 2',8-Anhydro-Adenosine 3A). This is neither a sentence nor a title. Proper nomenclature would be 2',8-anhydroadenosine (one word, lower case, no hyphen). Also, note the typographical error (presence)

Response: We have corrected this nomenclature and the typo.

Comment: At least in standard scientific English, 'Spectrum to show the ... products' should be 'spectrum showing the products of .. '

Response: We have changed “spectrum to show” to “spectrum showing”.

Comment: When the authors write 'RT', they probably mean room temperature (r.t.) not gas constant times absolute temperature (RT).

Response: Correct. We have changed “RT” to “room temperature”.

Comment: What is a 'lyophilite'?

Response: Lyophilite (or lyophilisate) is the material acquired upon lyophilisation (or freeze-drying). We have changed "lyophilite" to "lyophilisate", which is more widely used, and thank reviewer 3# for drawing this to our attention.

Comment: What is 'Pyrimidine UV Irradiation'? (Irradiation of pyrimidines?) What are 'Pyrimidine Oxidations'?

Response: We have changed the subtitles as suggested.

Comment: Sometimes it is not clear what was done. For example, '...white solid. These solids were dissolved in boiling H₂O (40.0 mL) and allowed to cool to RT. After 24 h precipitation of crystalline solid was observed. These solids were isolated by filtration to afford 31.2 mg of 2',8-anhydro-inosine 3I (31%) as off-white translucent crystals.' (Solids that are dissolved cannot be allowed to cool because they are no longer solids. Also, the switching between singular and plural makes this section more than confusing. It sounds as if several fractions [batches] were combined)

Response: We have corrected this statement. Specifying that: **a)** "...and the resultant solution was allowed to cool". **b)** "After 24 h a crystalline precipitate was observed. The precipitate was isolated..."

Comment: The general part and some figure legends give the pulse program for the acquisition of one-dimensional proton NMR spectra as "noesygppr1d". Why is this pulse program being used? The title suggests that it is for NOESY spectra, not for 1D spectra (typically called "zg" in Bruker parlance).

Response: Noesygppr1d is a standard Bruker 1D nuclear Overhauser enhancement spectroscopy pulse sequence with presaturation during relaxation delay. We believe the last two characters (i.e. 1d) denote this is a 1D NMR technique. Noesygppr1d is a well-established and widely used 1D NMR technique, which can be found in Bruker pulse sequence catalogues and literature describing sample quantification by NMR spectroscopy (e.g.

https://www.bruker.com/fileadmin/user_upload/8-PDF-Docs/MagneticResonance/NMR/brochures/Detergent_apps_note_T150217.pdf).

This method suppresses the solvent (water) at the beginning of the pulse sequence and produces pure phase signals minimally affected by the relaxation. Noesygppr1d is one of a number of available water suppression techniques, however noesygppr1d represents one of the optimum choices for obtaining a good resolution spectra for samples containing multiple small molecules. Accordingly noesygppr1d is the standard choice for quantification of molecules in biofluids and tissue samples, and in metabolic profiling. It is also both well suited to our requirements and well preceded in the literature [e.g. Beckonert et al *Nat. Protoc.* **2**, 2692–2703 (2007); Mckay *Concepts Magn. Reson. Part A Bridg. Educ. Res.* **38A**, 197–220 (2011); Panteleimon et al *Nat. Commun.* **8**, 1662 (2017); Stairs et al *Nat. Commun.* **8**, 15270 (2017); Islam et al *Nat. Chem.* **9**, 584–589 (2017)].

Comment: Why are NMR data listed for known compounds? ... repeating data from the literature is a waste of space, even in the SI.

Response: We thank reviewer #3 for this suggestion. We have removed all superfluous data.